# Predicting the Amount of Electric Power Transaction Using Deep Learning Methods

**Gwiman Bak [1]** and **Youngchul Bae [2],***

[1] Department of Electrical and Semiconductor, Chonnam National University, Yeosu 59626, Korea; qkrrlend@naver.com

[2] Division of Electrical, Electronic Communication and Computer Engineering, Chonnam National University, Yeosu 59626, Korea

* Correspondence: ycbae@chonnam.ac.kr; Tel.: +82-10-8996-6839

**Abstract:** The most important thing to operate a power system is that the power supply should be close to the power demand. In order to predict the amount of electric power transaction (EPT), it is important to choose and decide the variable and its starting date. In this paper, variables that could be acquired one the starting day of prediction were chosen. This paper designated date, temperature and special day as variables to predict the amount of EPT of the Korea Electric Power company. This paper also used temperature data from a year ago to predict the next year. To do this, we proposed single deep learning algorithms and hybrid deep learning algorithms. The former included multi-layer perceptron (MLP), convolution neural network (CNN), long short-term memory (LSTM), gated recurrent unit (GRU), support vector machine regression (SVR), and adaptive network-based fuzzy inference system (ANFIS). The latter included LSTM + CNN and CNN + LSTM. We then confirmed the improvement of accuracy for prediction using pre-processed variables compared to original variables We also assigned two years of test data during 2017–2018 as variable data to measure high prediction accuracy. We then selected a high-accuracy algorithm after measuring root mean square error (RMSE) and mean absolute percent error (MAPE). Finally, we predicted the amount of EPT in 2018 and then measured the error for each proposed algorithm. With these acquired error data, we obtained a model for predicting the amount of EPT with a high accuracy.

**Keywords:** Korea electric power transaction; short-term load forecasting; prediction; power transaction; deep learning

## 1. Introduction

Nowadays, power consumption is gradually increasing due to rising introduction of smart factories, electric cars, and embedded systems by adapting the concept of automatization, unmanned plant, and artificial intelligence in the industry. As power consumption increases, more efficient and stable operation of the power system is needed. One of the most important things to operate power systems is that the supply and the demand of electric power should fit the balance within a fixed range of electricity reservation ratio. Generally, when the power supply is larger than the power demand, then surplus power is generated which is uneconomic. When the power supply is smaller than the power demand, then a power failure occurs, which may cause a blackout.

In South Korea, Korea Electric Power Company (KEPCO) was responsible for electric power production (generation), transportation (transmission and substation), and selling (distribution) before year 2000. Thus, electric power trading as an integrated management between power supply and power demand was not important for KEPCO. However, KEPCO was separated into electric power production, transportation, and selling after year 2000. As the power plant responsible for electric power



production is split into several subsidiary companies, the establishment of Korea Power Exchange (KPX) is necessary. Thus, the role and importance of KPX for electric power trading are increasing.

In order to keep the balance between supply and demand of electric power, the electric power exchanger need to figure out electric power demand accurately. When the exact electric power demand is identified, power exchange companies can perform stable power trading with individuals and companies that possess nuclear power, hydroelectric power, thermal power, and renewable energy such as solar photovoltaic power and wind power generation. Thus, companies and individuals can lay down various schemes necessary for electric power production. In addition, the KPX is known as a joint-stock company that maximizes profit on its business. It needs stable electric power trading and optimized operation planning. To do this, accurate prediction of its electric power trading amount is required. If its prediction of the amount of EPT such as demand and supply of the electric power is not accurate, then two problems can occur. First, if predicted electric demand is more than the electric supply amount, then generation cost rises because surplus electric power can happen with overuse of generation facilities. Second, if predicted electric demand is less than electric supply amount, then the lack of electricity reserve can happen, causing a blackout. These two items can increase instability of electric power systems. In South Korea, for example, the KPX failed to predict electric power demand or consumption on 15 September 2011. As a result, they experienced a power outage. Since then, the importance of predicting the amount of EPT has been continuously highlighted. Thus, research on prediction or forecast of electric power demand is increasing.

Yang et al. (2020) have reported that the number of publications on the prediction of electric power demand or consumption has been steadily increasing for 20 years, from eight in 1999 to 148 in 2018 [1]. Power demand forecasting for the amount of EPT needs to be predicted by time, day, month, year, and so on because the predicted value can be different according to time scale. There are deep correlations among electric power demand, amount of EPT, and optimal power generation. An optimal power generation plan of the plant and the purchase or sale of exchange systems might have a direct influence. Generally, prediction of electric power demand can be classified as Very Short-Term Load Forecasting (VSTLF) which extends from seconds to minutes to minimize network response to demand flow, short-term load forecasting (STLF) which minimizes day-to-day planning and shipping costs, medium-term load forecasting (MTLF) which plans to operate power generation, and long-term load forecasting (LTLF) which plans to expand network [2]. Criteria, range of applying time, and aim for the classification of electrical power demand are summarized in Table 1.

**Table 1.** Classification of electricity demand by four criteria.

| Criteria | Input Variables | Aim |
|---|---|---|
| Long-term load forecasting (LTLF) | Range of month | Expansion planning of the network |
| Medium-term load forecasting (MTLF) | Range of weeks | Operational planning |
| Short-term load forecasting (STLF) | Range of day | Day of day planning and dispatch cost minimization |
| Very short-term load forecasting (VSTLF) | Minutes or hours | Scale of seconds to minutes allows the network to respond to the flow of demand |

Among various techniques for predicting electric power demand, STLF is an essential component of energy management systems (EMS) because it provides input data for load flow and accidental analysis [3]. Hernandez et al. (2014) [4] have explained that weekly, daily, and hourly forecasts are the most important forecasts. Among the four forecasts they emphasize prediction of power demand for the next 24 h because power companies require accurate forecasting power demand.

Power transaction represents actual transactions between power generation and KEPCO in the power market. Power transactions have the same characteristics as power demand. Thus, in this paper, we replaced demand for electricity with amount of EPT. In South Korea, the KPX performs power trading.

The KPX as the source of Korea's EPT has the following characteristics:

- The KPX operates the electricity market under a cost-based pool (CBP) system which determines prices based on actual costs.
- The power industry structure is independent only for the power generation sector while and sectors of transmission and distribution sales are operated exclusively by KEPCO.
- Except for power generation companies that have signed a separate power purchase agreement with KEPCO, all electric power generated by generators with capacity of 20 MW or more must be traded through the Korea Electric Power Exchange.

As shown in Figure 1, the KPX maintains an energy system which has a vertical integrated monopoly structure except for the power generation sector. Thus, it is possible to collect nationwide data.

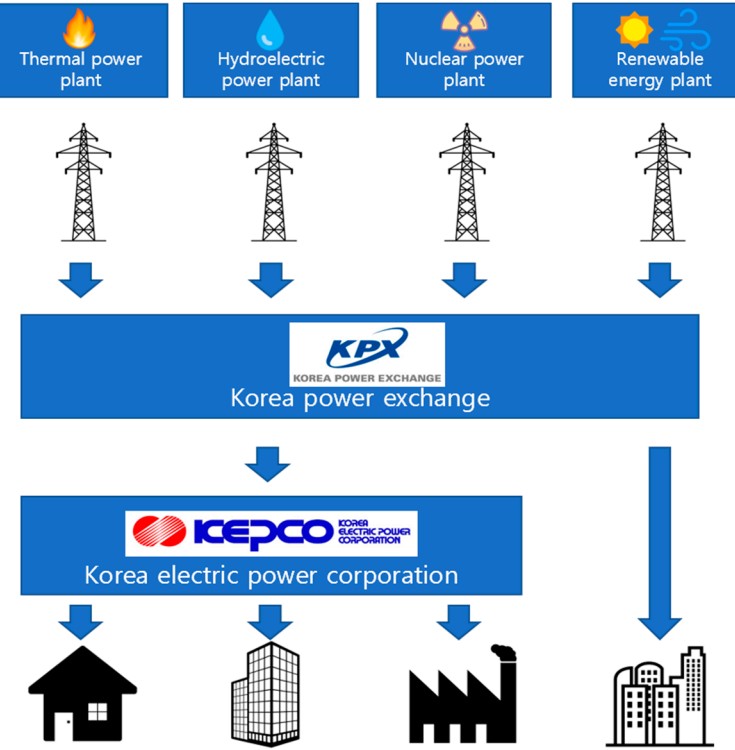

**Figure 1.** Korea energy system [5,6].

Previous studies about the amount of EPT have achieved successes for prediction with low accuracy. They used variables such as temperature, growth domestic product (GDP), holiday, sunlight and humidity to predict the amount of EPT, which will occur after starting day of prediction. Therefore, each variable are also necessary to predict the prediction error occurred in each variable. For such reason, previous studies about the prediction of the amount of EPT have a critical point for long-term prediction with one year or more. To overcome such problem, we need to choose variables that can be acquired on the starting day of prediction to predict the amount of EPT.

In this paper, we choose variables which occurred before starting day of prediction. The amount of EPT was then predicted using deep learning. To do this, we proposes single deep learning algorithms including MLP, CNN, LSTM, GRU, SVR, and ANFIS and hybrid deep learning algorithms including LSTM + CNN and CNN + LSTM to predict the amount of daily EPT STLF in KPX of South Korea. We then confirmed the improvement of accuracy for prediction using pre-processed variables compared to original variables. We also assigned two years of test data during 2017–2018 as variable data to measure prediction accuracy. We then selected an algorithm with high accuracy after measuring RMSE and MAPE.

Key processes used in this study are as follows.

- Variables were selected by investigating characteristics of EPT in South Korea.
- Data of selected variables were obtained from one year ago at the time of EPT in KPX of South Korea.
- Variables were pre-processed to increase their correlations with the amount of EPT.
- Various algorithms such as single algorithms including CNN, LSTM, GRU, and SVR and hybrid algorithms including CNN + LSTM and LSTM + CNN were applied.

The organization of this paper is as follows.

Chapter 2 introduces deep learning algorithm and prediction for EPT and demand. Chapter 3 explains features of EPT amount in South Korea and describes correlations between characteristics of EPT amount and features for variables used. Chapter 4 describes the preprocessing process of variables to increase the prediction performance of deep learning. Chapter 5 presents the process of predicting the amount of EPT using deep learning, introduces each algorithm, and designates parameters of each algorithm through an empirical process. Chapter 6 analyzes prediction values of each algorithm by specific pattern and measures the error of prediction value for one year. Finally, Chapter 7 shows results of prediction for EPT amount and proposes directions for future research.

## 2. Related Works

Research for predicting the amount of EPT has been studied such as VSTLF, STLF, MTLF, LTLF. Many researchers have predicted the amount of EPT using time series [7], fuzzy theory [8], and neural network [9]. Prediction techniques using artificial intelligence including deep learning and machine learning have also been reported recently.

Artificial intelligence including deep learning and machine learning has been widely researched in various fields such as autonomous driving vehicle [10], global horizontal irradiance [11], stock prices [12], wind speed [13], traffic flow [14], and prediction of EPT amount and demand. Related research studies on the amount of EPT are summarized below.

González-Romera et al. [15] have predicted monthly electricity transaction volume energy demand using an MLP algorithm for market research and maintenance plan of electricity producers. They used two methods to predict the volume of electrical power demand: trend of time series and monthly fluctuation. However, their paper has a critical point in that only long-term prediction is possible.

Lin et al. [16] have classified users according to the type of electricity consumption and predicted power consumption amount of each group using an LSTM algorithm. Their method showed better prediction performance than conventional multiple linear regression based on error of MAPE and $R^2$.

KASULE and AYAN [17] have applied an ANFIS algorithm to predict power consumption in Uganda and presented a long-term power prediction model for easier market prediction. They used Particle Swarm Optimization-ANFIS (PSO-ANFIS) and Genetic Algorithm-ANFIS(GA-ANFIS) to optimize parameters of the model. The prediction performance of their method was better than the multivariate linear regression model. Ogihara et al. [18] have predicted the amount of electricity power demand in Japan. They used a multiple stress model and Artificial Neural Network (ANN) as algorithms for prediction. To do this, they eliminated holidays and performed forecasting using two weeks, one month, and two months, respectively. They also used economic indicators such as opening price, high price, low price, and volume as variables in order to achieve prediction with high accuracy. Predictable economic indicators would be very useful variables as very good references to predict power demand. However, they did not describe these values as variables.

Duong-Ngoc et al. [19] have predicted electricity demand (per hour) over a week in Ho Chi Minh City, Vietnam. They used feed-forward deep neural network (FF-DNN) and recurrent deep neural network (R-DNN) algorithms. They also used temperature, holiday, and electric power demands before day, hour, and week as variables. They showed a high accuracy by using features of the algorithm. However, it was impossible to predict in the long term because the amount of electricity demand was predicted using variables a day ago and an hour ago.

Eshragh et al. [20] have predicted electricity demand (weekly) in New South Wales (NSW), South Australia (SA) and Victoria (VIC) regions of Australia. To do this, they developed a hybrid

algorithm with seasonal auto regressive integrated moving average (SARIMA) and a linear model. They used the lowest temperature, the highest temperature, and amount of sunlight on the day of demand for electricity as variables. They showed that the proposed algorithm had a higher accuracy than the recurrent neural network (RNN) algorithm.

Kim et al. [21] have profiled Korea's temperature, humidity, holidays, day of the week, and n-day of the season as variables and showed a high-accuracy prediction of power demand. They used a hybrid model by combining advantages of LSTM and CNN. Considering electric power demand value as a key value and other variables as contest information, data were preprocessed as a <key, contest > pair. Through this process, important contextual information for training neural networks was efficiently used.

Del real et al. [22] have predicted power demand in France by combining CNN and ANN commonly used for image classification and showed a higher accuracy than Support Vector Machine (SVM), ARIMA, and ANN. Variables used in their study included temperature of the day before the power demand forecast day, week (1 to 52), hour (0 to 23), day of the week (0 to 6), and public holidays (false true).

Li et al. [23] have predicted electricity demand in Australia and Singapore using algorithms such as SVM and extreme learning machine (ELN). In order to increase accuracy, they extracted data noise. They also used fast Fourier transform (FFT) in order to grasp and remove periodicity of power demand.

Imani [24] has provided an improvement of forecasting performance by selecting an appropriate feature space for Iran's electricity demand. The provided domain contained complementary information about the shape and variation of electrical load sequence. Obtained load characteristics were then integrated with original load values in the time domain and input to the LSTM.

Ma et al. [25] have forecasted energy consumption in South Africa for 2017–2030. They used ARIMA and Nonlinear Gray Model-Autoregressive Integrated Moving Average (NGM-ARIMA) algorithms with energy consumption in 1998–2016 as variables for the prediction.

Table 2 summarizes recent studies on the prediction of EPT amount for each country.

**Table 2.** Previous studies on the prediction of amount of EPT.

| Number | Reference | Year | Country | Model | Description |
|--------|-----------|------|---------|-------|-------------|
| 1 | [18] | 2019 | Japan | MLR, MLP | • Uses meteorological data,<br>• Optimizes combinations when predicted values select<br>• Organize seasonal models<br>• Predicted start point is older than used variable |
| 2 | [19] | 2019 | Vietnam | FF-DNN, RNN | • Variables are used one day ago, one hour ago and one week ago.<br>• Use as a variable that cannot be acquired at the beginning of the prediction |
| 3 | [20] | 2020 | Australia | SARIMA | • Develop a SARIMA-regression model for the weekly power demand<br>• Use as a variable that cannot be acquired at the beginning of the prediction |
| 4 | [21] | 2019 | Korea | LSTM + CNN | • Pre-process data by pairing for power demand values.<br>• Predicted start point is older than used variable |

**Table 2.** *Cont.*

| Number | Reference | Year | Country | Model | Description |
|---|---|---|---|---|---|
| 5 | [22] | 2020 | French | CNN + ANN | • Using a combination of CNN and ANN<br>• Predicted start point is older than used variable |
| 6 | [23] | 2020 | Australia and Singapore | SVR, sine, cosine optimization algorithm et. | • Combining SVR, sine and cosine optimization<br>• Data preprocessing method was used (AFD, FFT)<br>• Predicted start point is older than used variable |
| 7 | [24] | 2020 | Iran | LSTM | • Significantly improve electric load forecasting by integrating acquired load characteristics and time series<br>• Predicted start point is older than used variable |
| 8 | [25] | 2020 | South Africa | ARIMA, NGM-ARIMA | • On the basis of NGM and ARIMA single models, a new combined NGM-ARIMA model proposed<br>• Predicted start point is older than used variable |
| Propose method | - | - | Korea | CNN, LSTM + CNN, GRU etc. | • The predicted start point is placed in the future for variable used<br>• Preprocess variables for better prediction of power demand |

Except for one study [25], papers presented in Table 2 performed predictions by selecting various algorithms and various variables. Although variables used in these studies should not be used for the prediction starting date, these variables were used on the after prediction date. Thus, there are limitations for long-term prediction when variables used are based on the after prediction date. In Figure 2, 2-1 represents used variables proposed in this paper and 2-2 represents used variables in the previous work. The point of 2-1 has no error because we knew the variable value. On the other hand, the point 2-2 shows prediction error because it required prediction.

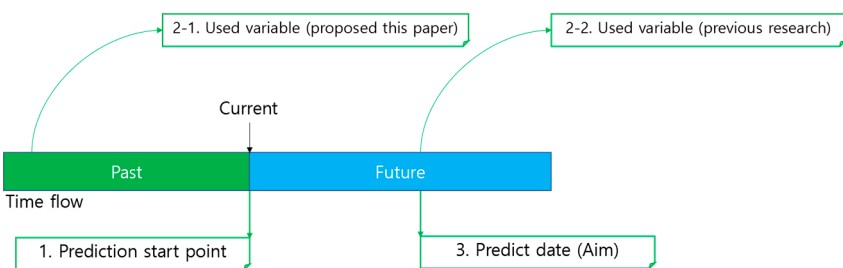

**Figure 2.** Time point of selected variable.

This paper attempts to predict the amount of EPT by selecting the point of variable 2-1 that can be used on the starting date of prediction for selecting various algorithms and parameters.

## 3. Data Characteristics

In this section, data characteristics such as the amount of EPT, temperature, and special day are reviewed.

### 3.1. Amount of EPT of Korea Power Exchange

In order to predict the amount of EPT, first of all, it is necessary to apprehend the pattern of the amount of EPT. The amount of EPT in South Korea largely has three characteristics: year periodicity, weekday periodicity, and pattern of special day. First, South Korea has four distinct seasons of spring, summer, fall and winter. According to seasonal changes, climate including temperature and humidity has year-round periodic characteristics. Second, due to the amount of EPT which is closely related to people's lives, according to the day of the week, the pattern of the amount of EPT has a periodicity. For example, every Monday or Tuesday has nearly the same amount of power transaction. Third, EPT in South Korea has a special day pattern. For holidays including New Year holiday (both solar and lunar calendars), Thanksgiving Day, and two, three, or more days of holiday, irregular pattern of EPT can be observed depending on weather, day of the week, and business condition.

Figure 3 presents the amount EPT from 1 January 2014 to 31 December 2018.

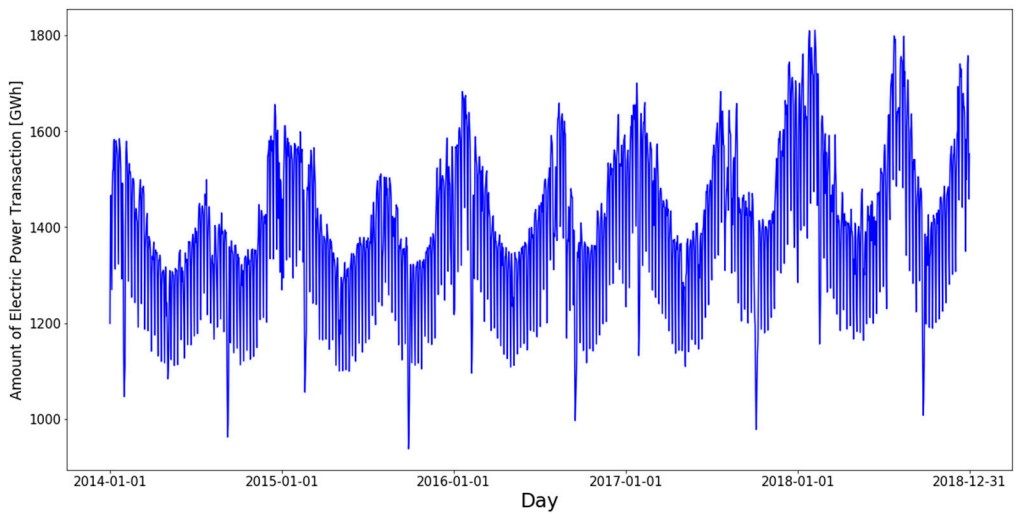

**Figure 3.** Amount of EPT in Korea from 2014 to 2018.

Figure 4 presents the amount of EPT from January 2017 to December 2018 in South Korea, showing annual periodicity, weekly periodicity, and special day patterns.

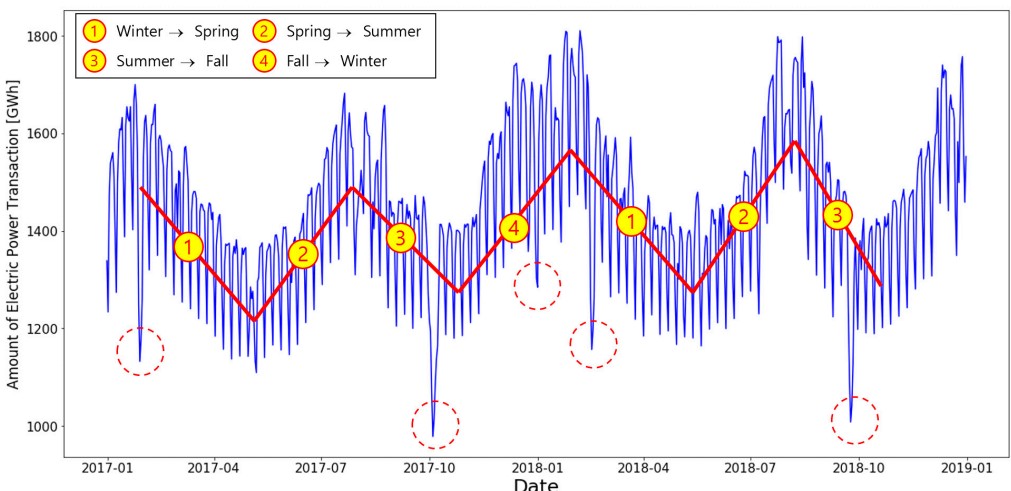

**Figure 4.** Characteristics of EPT from 2017 to 2018.

Red lines indicate annual periodicity of the amount of EPT. Lines that frequently and repeatable rise and fall indicate weekly periodicity.

Numbers ①, ②, ③, and ④ indicate transition status from winter to spring, from spring to summer, from summer to autumn, and from winter to autumn, respectively.

Red circle shows the pattern that occurs when it is a holiday (New Year's Day, Thanksgiving Day) in a row during a special day.

Figure 5 shows geographical locations of Seoul, Gwangju, and Busan in South Korea.

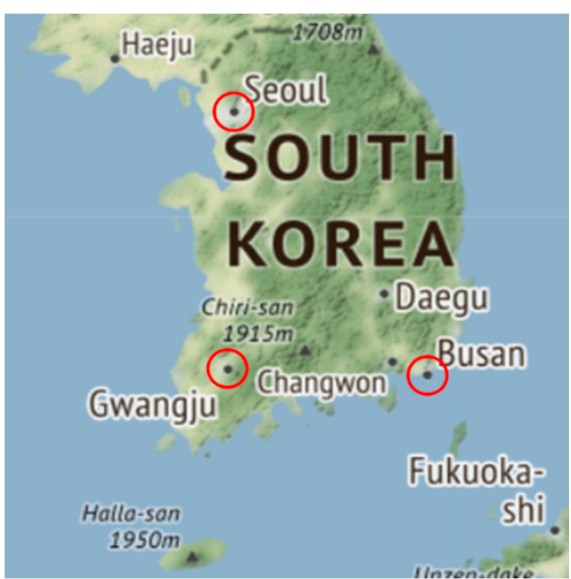

**Figure 5.** Geographical locations of Seoul, Gwangju, and Busan in South Korea.

*3.2. Temperature*

The temperature of South Korea has characteristics of four distinct seasons. Rainy season and typhoon occur mainly during the summer season, although global warming is affecting the time of them recently. Figure 6 shows the amount of EPT and the highest temperatures in Seoul, Busan, and Gwangju from 2014 to 2018. Here, the blue line represents the amount of EPT and red, green, and yellow lines represent Gwangju, Busan, and Seoul's highest temperatures, respectively.

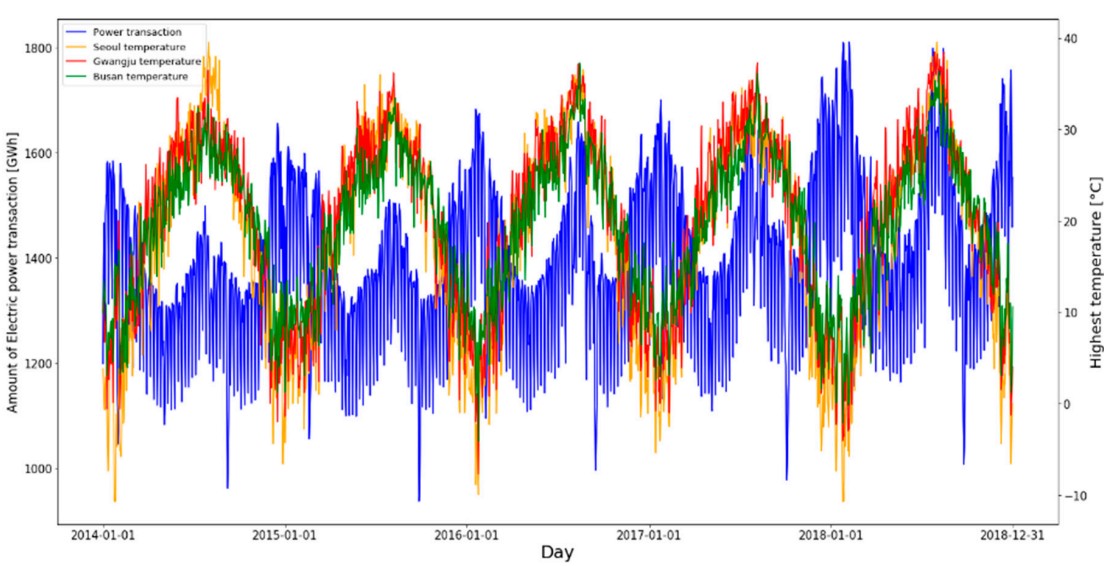

**Figure 6.** Amount of EPT in South Korea and the highest temperature in Seoul.

In order to use temperature as a variable based on the prediction date, we obtained the highest temperatures of day in Seoul, Gwangju, and Busan from 1 January 2013 to 31 December 2017 one year ago based on the amount of EPT from 1 January 2014 to 31 December 2018.

Figure 7 presents the highest temperatures of day in Seoul, Gwangju, and Busan from 1 January 2013 to 31 December 2017.

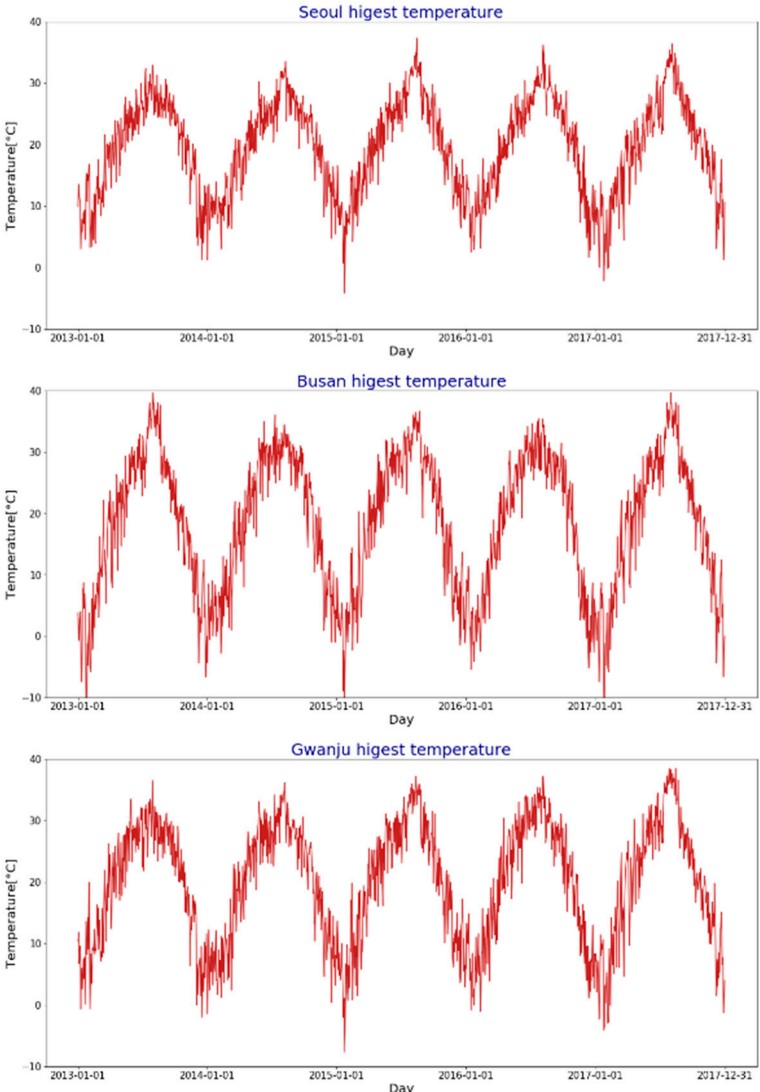

**Figure 7.** Highest temperatures of day in Seoul, Gwangju and Busan during 2013–2017.

The highest and the lowest temperatures of year in South Korea were observed to in August of the summer season and January of the winter season, respectively. Temperature features showed periodic characteristics of seasonal changes. Temperatures ascend rapidly in spring season until early summer. It means that from early summer, the rising of temperature is getting smaller. Temperatures then descend rapidly after August, falling into lower than the freezing point in the winter season.

EPT changes according to seasonal variation, showing periodic characteristics of a year that reflect four distinct seasons. Generally, temperatures affect demands for cooling and heating. Thus, high and low temperatures will increase or descries the demand for electricity.

Table 3 summarizes correlation between amount of EPT and temperature.

**Table 3.** Correlation between amount of EPT and temperature.

| Number | Sortation | Temperature | Amount of EPT | Correlation |
|:---:|:---:|:---:|:---:|:---:|
| ① | Winter → Spring | Increase | Decrease | − |
| ② | Spring → Summer | Increase | Increase | + |
| ③ | Summer → Fall | Decrease | Decrease | + |
| ④ | Fall → Winter | Decrease | Increase | − |

The numbers in Table 3 represent the numbers of sections in Figure 3.

Numbers ①, ②, ③, and ④ indicate transition status from winter to spring, from spring to summer, from summer to autumn, and from winter to autumn, respectively.

In Section ① and Section ④, when the temperature was increasing, the amount of EPT was decreasing. Additionally, the amount of EPT was increasing when the temperature was decreasing. Amount of EPT for Section ② and Section ③ were also increasing (or decreasing) when the amount of EPT was decreasing (or increasing).

The solar calendar has a leap day in quadrennial. Data of temperature, amount of power transaction, etc., were acquired during 2013–2017. A leap occurred in 2016. In order to predict the amount of power transactions on 29 February 2016, the highest temperature variable of each city on 29 February 2015 must be generated. The maximum highest temperature variable for 29 February in 2015 was generated by averaging the highest temperatures of each city on 28 February and 1 March 2015.

### 3.3. Special Day

Since the amount of EPT is closely related to how people live, daily EPT during the week also shows periodicity affected by holidays. Table 4 indicates current public holidays in South Korea.

**Table 4.** Features of Korean holidays.

| Public Holiday | Date | Solar/Lunar | Replacement Holiday |
|:---:|:---:|:---:|:---:|
| New year's day | 1.1 | S | |
| Lunar New Year's Day | 12.31–1.2 | L | √ |
| Independence Movement Day | 3. 1 | S | |
| Buddha's birthday | 4. 8 | L | √ |
| Children's Day | 5. 5 | S | |
| Memorial Day | 6.6 | S | |
| National Liberation Day | 8.15 | S | |
| Thanksgiving Day (Chuseok) | 8.14–8.16 | L | √ |
| the National foundation Day of Korea | 10.3 | S | |
| Hangul Proclamation Day | 10.9 | S | |
| Christmas | 12.25 | S | |
| Election Day | | | |

There are solar and lunar holidays in South Korea. The solar calendar is made based on the sun. It has year-based dates that revolve around the sun. The lunar calendar is dated based on the moon as center on the period while the moon goes around the earth. Solar holidays are fixed even the year changes. Lunar holiday changes by the date of solar calendar every year.

When holidays such as lunar New Year's Day and Thanksgiving Day are overlapped with Sunday, or Children's Day overlaps either on Saturday or Sunday, alternative holidays are designated as holidays at the first coming days of non-holiday. The most popular social special day is the election day in South Korea. The presidential election is held every five years. The parliamentary election is held every four years. National simultaneous local elections are held every four years in South Korea.

Figure 8 indicates Election Day interval from 2014–2018. The light green graph represents amount of EPT. The red vertical line represents the Election Day. The blue vertical line represents Saturday. The green vertical line represents Sunday. The declining pattern of the amount EPT for Election days was similar to the pattern of holiday.

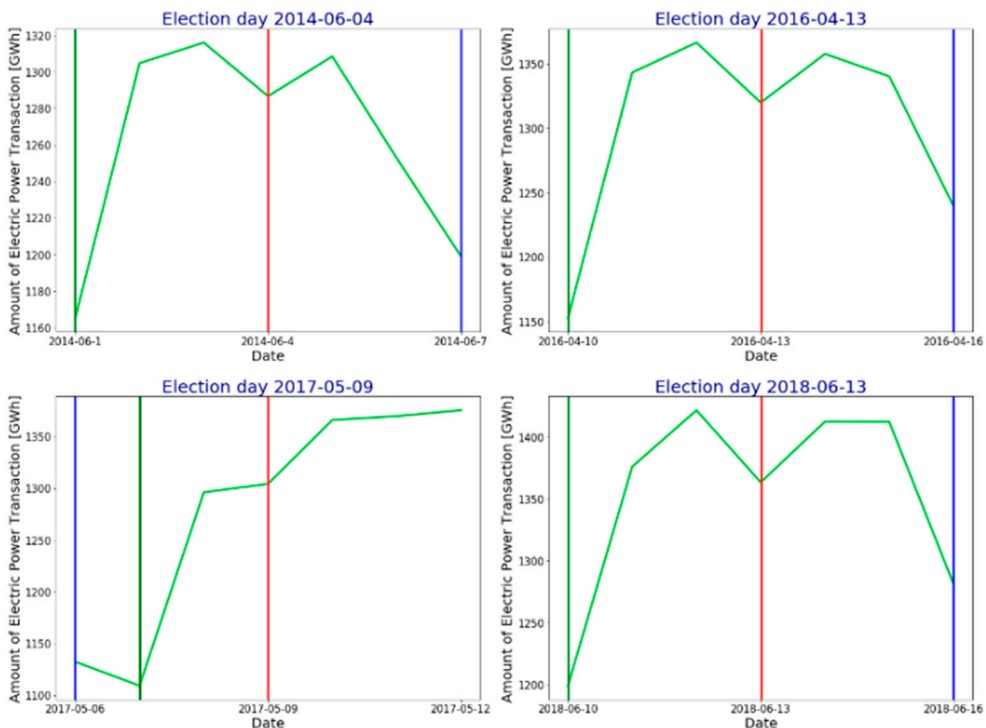

**Figure 8.** Election Days in South Korea during 2014–2018.

Figure 9 depicts Children's Day in South Korea from year 2014–2018.

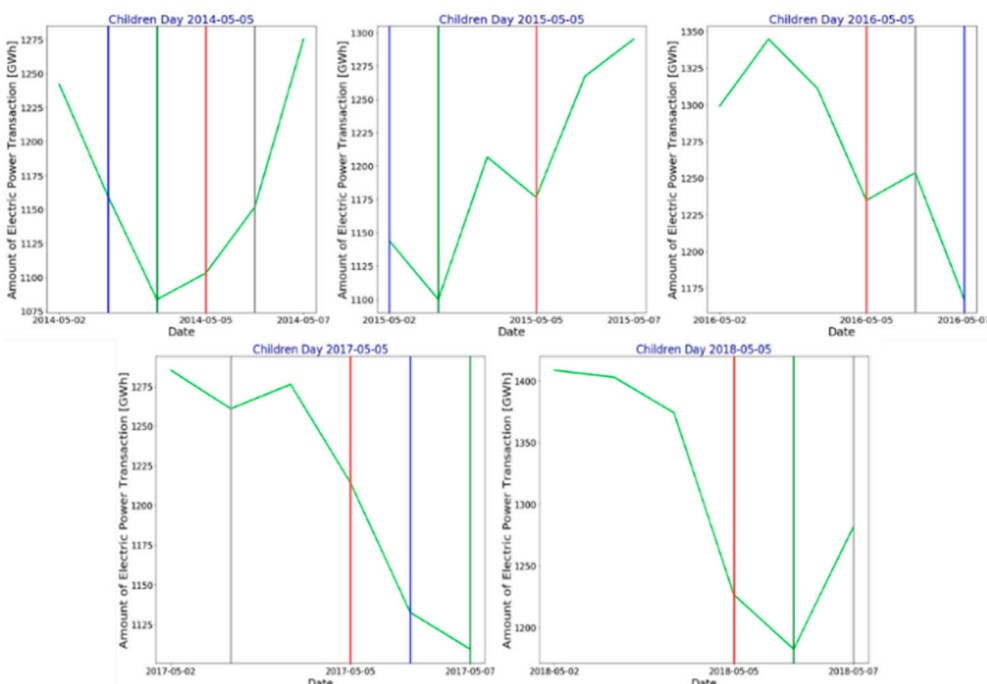

**Figure 9.** Children's Day in South Korea during 2014–2018.

In Figure 9, the light green graph represents amount of EPT. Red vertical, blue vertical, green vertical, and gray vertical lines represent Children's day, Saturday, Sunday, and alternative holiday or Buddha's day, respectively. Amount of EPT was decreased on these days as seen on Election Days.

The pattern of Saturday and Sunday showed that the amount of EPT changed with a constant pattern within almost the same range. The amount of EPT was constantly declining during intervals of Friday–Saturday and Saturday–Sunday. The amount of EPT decreased from Friday to Sunday, although it began to increase during the interval of Sunday–Monday followed by a stabilized amount until Friday. This can be seen as a phenomenon that industrial loads is reduced because most industrial actions are stopped during weekends and start to operate again after work hours on Monday. Since holidays of Lunar New Year and Thanksgiving Day in South Korea are designated as holidays in lunar calendars, their dates are different every year.

Figures 10 and 11 show the beginning and the ending of holidays for lunar New Year holidays and Thanksgiving Day indicated by red vertical lines. The light green graph represents amount of EPT. Red vertical lines represent beginning and the ending of holidays

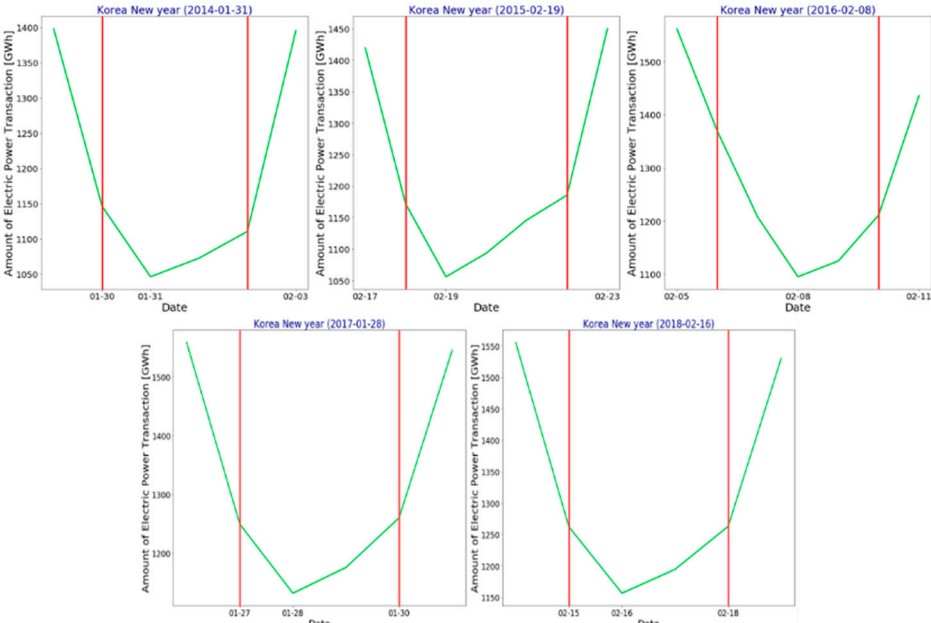

**Figure 10.** New year holidays in South Korea during 2014–2018.

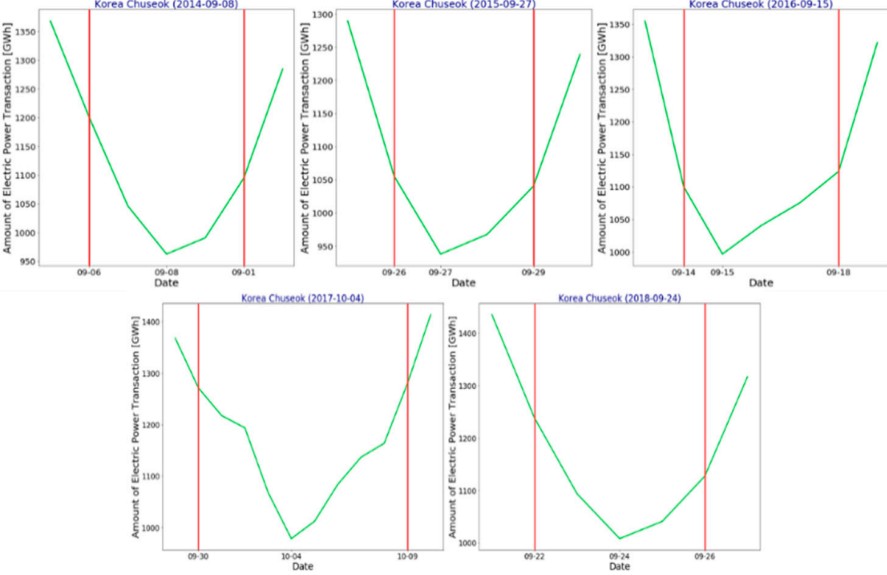

**Figure 11.** Thanksgiving holiday in South Korea during 2014–2018.

The holiday season is different every year. Thus, the holiday in a row includes lunar New Year holiday, alternative holidays, weekends, and so on. The lowest amount of EPT was recorded for lunar New Year's Day and Thanksgiving Day except for lunar New Year's Day in year 2014. The amount of EPT was also lower during weekends of holidays in a row.

Table 5 presents acquired data for the amount of EPT and temperatures from KPX and Korea meteorological administration.

**Table 5.** Characteristics of acquired data.

| Data | Sources | Time Unit | Data-Type | Measure |
|---|---|---|---|---|
| Amount of power transaction | Korea power exchange | day | float | GWh |
| Temperature | Korea meteorological administration | day | float | Celsius |
| Year, month, day | - | day | Date-time | - |
| Holiday | - | day | True/False | - |

Data for the amount of EPT were acquired by the KPX. Such data showed total amount of EPT in South Korea. The data type was "float" and the unit was GWh. Since the acquired amount of EPT was based on the national standard, temperature data to be acquired must also designate the nation's temperature as a variable. However, temperature data need too much computation if the entire Korean city is designated as a variable. Thus, three large cities, Seoul metropolitan area, Yeongnam region, and Honam region (Seoul, Gwangju, and Busan), were selected. Temperature data were acquired from the Korea Meteorological Administration. We used the unit of Celsius.

In South Korean government, legal holidays are designated according to "Regulations on Public Holidays of Public Offices" of Korea's Presidential Decree. In this paper, we assigned 1 to holiday and 0 to a business day as a true or false type.

## 4. Preprocessing Data

Typically, it difficult to predict the desired pattern when applying data as a variable without preprocessing data. Therefore, in this paper, we performed preprocessing of data in order to improve the prediction performance by applying artificial intelligence algorithms after acquiring data. The preprocessing of data was performed for temperature and amount of EPT during weekends and holiday in a row as special days.

### 4.1. Temperature

If the temperature is high or low, the amount of EPT is increasing because temperature affects cooling and heating demand. Generally, we should set criteria for cooling and heating. However, there is no recommended or unified proposal for indoor temperature in Korean public institutions. In this paper, we set heating was a positive number and cooling was a negative number by applying standardization of temperature value. Since cooling and heating have positive correlations with the amount of EPT, we should unify both cooling and heating with a positive number using square as represented in Equation (1):

$$T\_temperature = \left(\left(\frac{temp - temp_{mean}}{temp_{std}}\right)\right)^2 \tag{1}$$

Here, *temp* is the highest daily temperature in Seoul, Busan, and Gwangju of South Korea, $temp_{mean}$ is the average value of *temp*, and $temp_{std}$ is the standard deviation of *temp*.

Figure 12 presents the highest temperature in Seoul, Busan, and Gwangju as blue lines. The highest temperature of each city is then applied into Equation (1) to find *T_temperature* indicated by red lines.

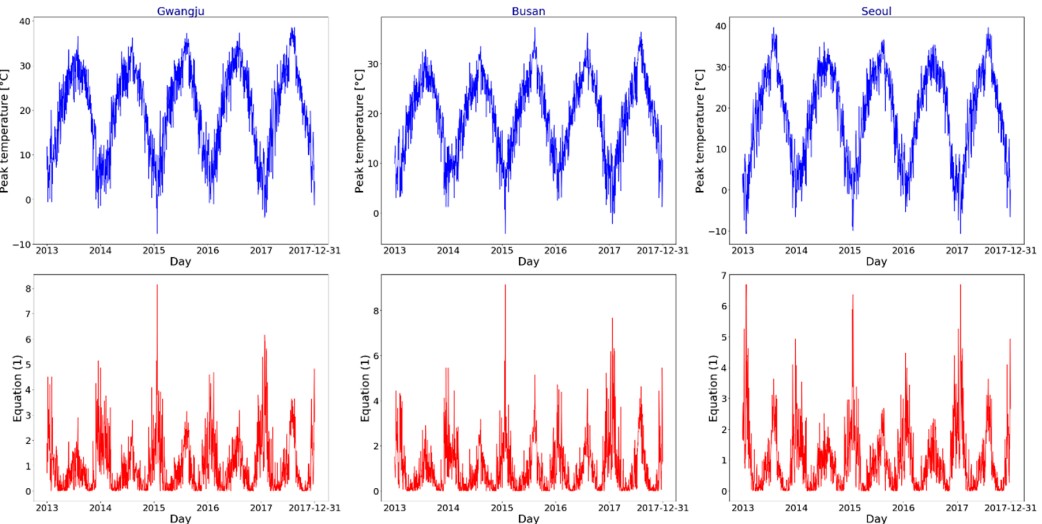

**Figure 12.** Graphs of temperature and graphs of results applied to Equation (1).

### 4.2. Special Day: Saturday, and Sunday

The pattern for the amount of EPT on weekdays showed a characteristic that it nearly linearly decreased it as the day passed on Saturday and Sunday. It then rapidly increased on Monday. To make variables reflect these patterns, holiday data were applied into Equation (2):

$$T\_holiday_1 = -(holiday + holiday_{-1}) \times holiday \qquad (2)$$

where *holiday* indicated whether it was a holiday and $holiday_{-1}$ indicated whether it was a holiday the day before. Both were binary.

Figure 13 shows different amounts of EPT in seven days and Equation (2) as a graph. Here, blue, red, and gray vertical lines represent Saturday, Sunday, and Monday, respectively.

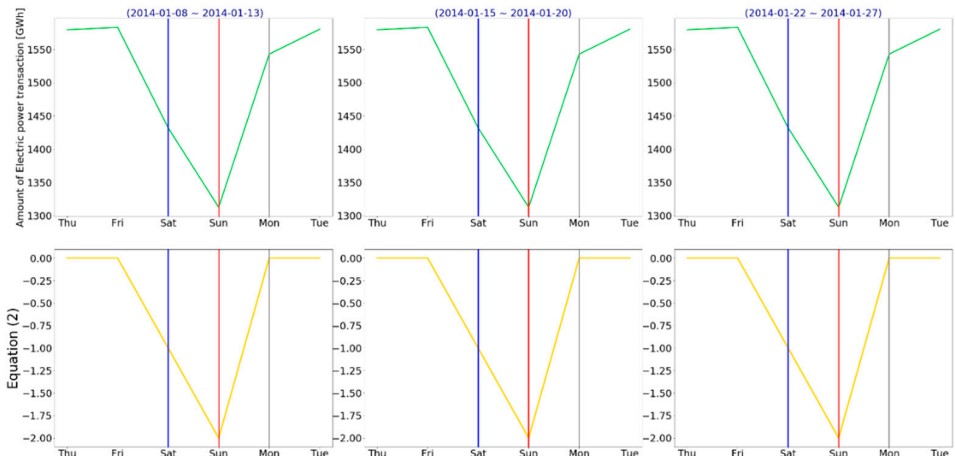

**Figure 13.** Graph applying power transaction and Equation (2) on different weeks.

### 4.3. Special Day: Continuous Holiday

The pattern for the amount of EPT on continuous holidays was different from the pattern of weekends. There were quite a few cases where the amount of EPT sharply dropped from the start day of the holiday in a row. The amount of EPT for continuous holidays was also significantly reduced compared to that for weekends. Thus, we need to create a pattern in which EPT decreases for continuous holidays. Such pattern is different from Equation (2). In addition, EPT on the start day

and the last day of the holiday should be lower than that of the rest of holidays while it should be the highest in the middle point of holidays.

Equation (3) refers to EPT in which patterns decrease during continuous holidays. It is characterized by adding up holidays between *n* days before and after the day of calculation so that the first day and the last day of holidays had lower EPT than the middle point.

$$T\_holiday_2 = -(\sum_{i=1}^{n} holiday_{-n} + holiday_n + holiday) \tag{3}$$

where *holiday* represents the holiday, $holiday_{-n}$ is a public holiday before n day, and $holiday_n$ is a public holiday after n day.

In this paper, in order to generate appropriate variables, we calculated correlation (*COR*) according to the *n* value. We then chose variables with the highest value among *COR* as shown in Equation (4):

$$COR = \frac{\frac{1}{n} \sum_{i=1}^{n} \left(F_i - \overline{F_i}\right)\left(O_i - \overline{O_i}\right)}{\sqrt{\frac{1}{n} \sum_{i=1}^{n} \left(F_i - \overline{F_i}\right)^2} \sqrt{\frac{1}{n} \sum_{i=1}^{n} \left(O_i - \overline{O_i}\right)^2}} \tag{4}$$

where $O_i$ is the observation value, $\overline{O_i}$ is the mean value of the observation value, $F_i$ is the prediction value, and $\overline{F_i}$ is the mean prediction value.

Table 6 presents *COR* ranked according to *n* value. As a result of comparison by *COR*, we obtained the highest variable *COR* which was the sum of three days (from the day before calculation to the day after calculation).

**Table 6.** *COR* rank by *n* value.

| n | COR | Rank |
|---|---|---|
| 3 | 3.2 | 1 |
| 5 | 0.2 | 5 |
| 7 | 2.1 | 2 |
| 9 | 0.4 | 4 |
| 11 | 1.2 | 3 |

## 5. Machine Learning Pipeline to Predict the Amount of EPT

The process of predicting the amount of EPT using deep learning is summarized in Figure 14.

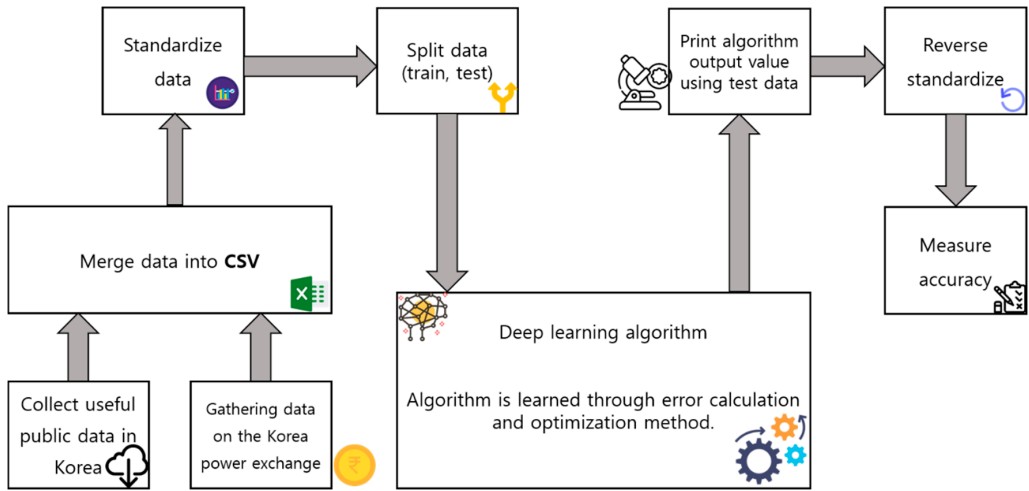

**Figure 14.** Algorithm used for predicting the amount of EPT.

First of all, to predict the amount of EPT, data for the amount of EPT were acquired from the KPX. Temperature data of Seoul, Busan, and Gwangju as a variable were obtained from the Korea Meteorological Administration as a comma-separated values (CSV) file. Second, acquired data for the amount of EPT amount, maximum temperature data, and holiday and daily data were merge into CSV type. Third, to analyze merged data as CSV type easily, we applied standardization. Standardize data were then divided into two categories: learning data and test data at a ratio of 70% to 30%. Fourth, the algorithm was trained using learning data. We also measured error and performed an optimization process. Then, we predicted the amount of EPT using test data. Finally, we carried out an inverse standardization for predicted results and measured the accuracy of prediction results.

We used total seven variables including years, months, days, holidays, and prior work. We also performed standardization for each variable in order to easily learn algorithms. Then we predicted the amount of EPT. Each variable used for prediction in this paper was placed in the past rather than at the starting point. We needed data acquired for years, months, days, and national holidays at any time. We could predict the highest temperature in each city because we used data a year ago.

Table 7 presents input data of years, months, days, and holidays as references for predicting the amount of EPT. We used holiday and temperature as preprocessing variables.

**Table 7.** Dates of variable used as a basis for forecasting the amount of EPT.

| EPT | Years | Months | Days | Holidays | Temperature |
|---|---|---|---|---|---|
| 2018-01-01 | 2018 | 1 | 1 | 1 | 2017-01-01 |
| 2018-01-02 | 2018 | 1 | 2 | 0 | 2017-01-02 |
| 2018-01-03 | 2018 | 1 | 3 | 0 | 2017-01-03 |
| 2018-12-30 | 2018 | 12 | 30 | 0 | 2017-12-30 |
| 2018-12-31 | 2018 | 12 | 31 | 0 | 2017-12-31 |

If we want to predict the amount of EPT on the date of 1 January 2018, we can use two kinds of data as variable. First, we can use data to appoint the day, for example the year, month, and public holidays in 1 January 2018. Second, we can use temperature data for 1 January 2017. We can acquire these variables as past data rather than data on the date of prediction.

There are four digits of data of year and the amount of EPT. However, there is maximum two digit for data of month, day, and holiday data. If we directly apply these data into the algorithm, there is a high possibility to have a large weight incline value and fall into local minimum. To reduce such possibility, we applied *standardization* as shown in Equation (5) which transformed zero as average into each variable and the amount of EPT.

$$standardization = \frac{x_i - x_{imean}}{x_{istd}} \tag{5}$$

where $x_i$ was data to be applied, $x_{imean}$ was the average value of $x_i$, and $x_{istd}$ was the standard deviation of $x_i$.

To measure the performance of each algorithm, we divided data into learning (or training) data and test data. Learning data accounted for 60% of data from 2014 to 2016 and test data accounted for 40% of data from 2017 to 2018. We trained the algorithm with learning data and measured the performance of algorithm with test data.

In this paper, we predicted the amount of EPT with eight algorithms and measured the error of test data. The measurement of error was performed using *RMSE* and *MAPE* as shown in Equations (6) and (7):

$$RMSE = \sqrt{\frac{1}{n} \sum_{i=1}^{n} (\alpha_i - \beta_i)^2} \tag{6}$$

$$MAPE = \frac{1}{n}\left(\sum_{i=1}^{n} \frac{|\alpha_i - \beta_i|}{|\alpha_i|}\right) \times 10 \tag{7}$$

where $n$ was the number of data, $\alpha_i$ was the amount of EPT, and $\beta_i$ was the predicted value.

Figure 15 presents the amount of EPT. Blue lines represent learning data and red lines represent test data.

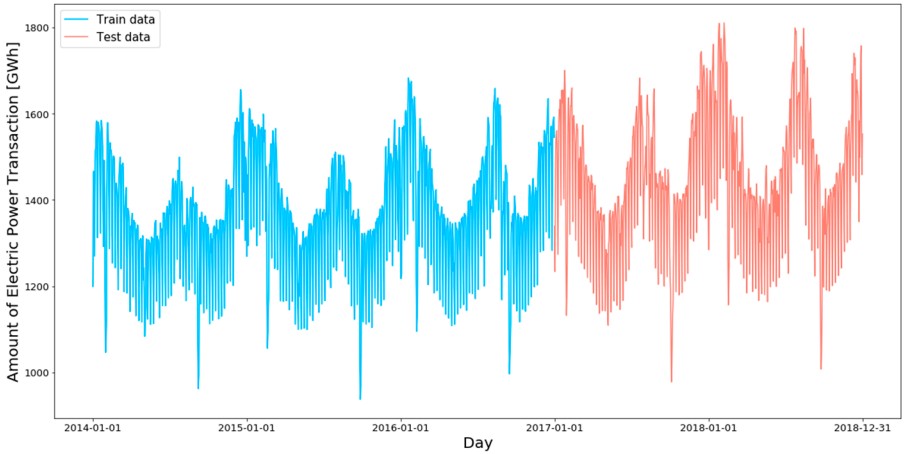

**Figure 15.** Composition between training data and test data.

In this paper, we proposed eight algorithms such as MLP, LSTM, CNN, GRU, SVR, ANFIS, CNN + LSTM, and LSTM + CNN to predict the amount of EPT. We also compared these results of prediction.

*5.1. MLP*

MLP is type of a sequence of several layers organized by perceptron. MLP consists of an input layer, a hidden layer, and an output layer. The input layer inputs the value of variable while the output layer gives the final output [26]. We performed training using learning data consisting of pairs of input and output. We determined the weight of connection line and deviation using information about certain values for the output when an input was given. The weight of connection line and deviation terms can be presented as shown in Equation (8). Figure 16 shows configuration of CNN, LSTM, and GRU algorithms based on MLP algorithms, not just MLP algorithms.

$$Y_{\mathrm{MLP}} = WX + b \tag{8}$$

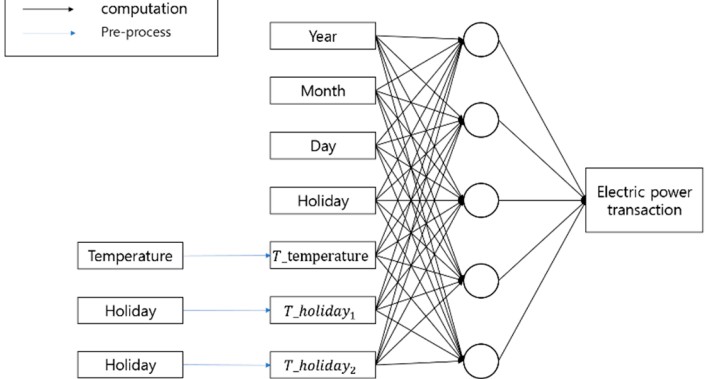

**Figure 16.** Configuration of MLP-based algorithm to predict the amount of EPT.

## 5.2. LSTM

LSTM is a derivative algorithm from RNN. The hidden layer of RNN consists of three gates: Input Gate, Output Gate, and Forget Gate [27]. Figure 17 shows the basic structure of LSTM.

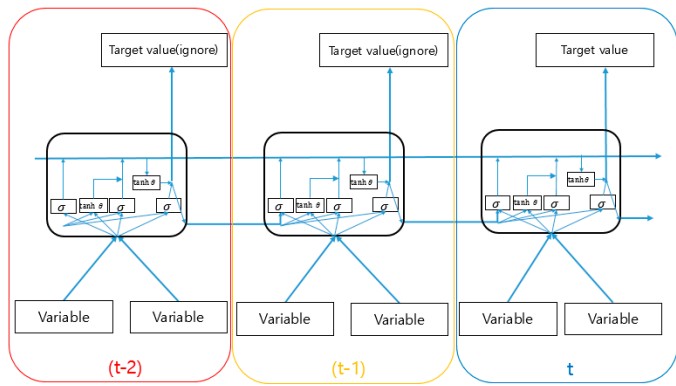

**Figure 17.** Basic structure of LSTM.

We added the following calculation processing. If (t-1) hidden layer of Figure 17 is changed into (t) hidden layer, (t-1) hidden layer will keep it or delete it vice versa.

Figure 18 presents the computational structure of the hidden layer of LSTM. The input variable is specified according to the time flow. The LSTM bundles variables, inputs variables into the hidden layer before N-day, and then outputs the current prediction value while ignoring the past prediction value. Figure 18 shows how we designate variables and target data.

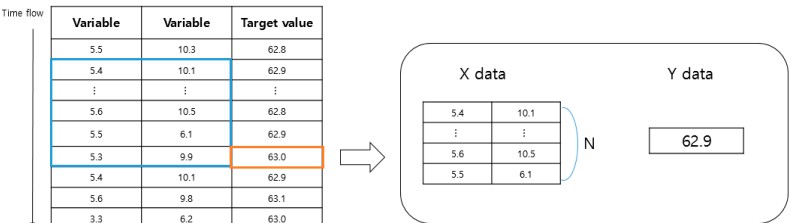

**Figure 18.** Designation of variables and target data.

As shown in Figure 18, after bundling the variable of time flow into (N) days, we designated it as input X data. We also designated Y output data as target value of t time of prediction target value. Table 8 and Figure 19 show RMSE values of train data and test data according to N and node values when nodes are 5.

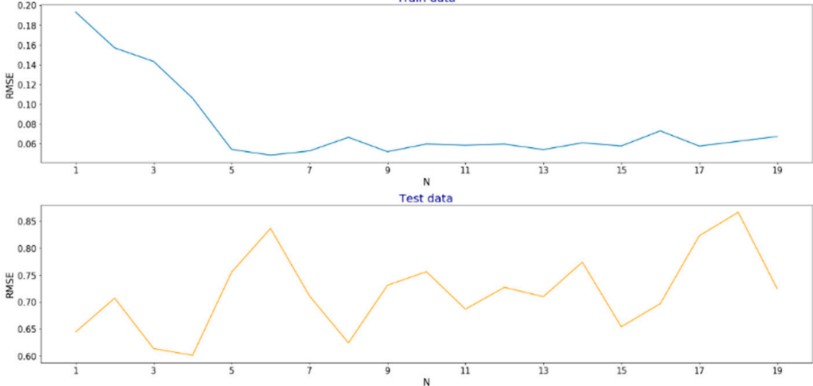

**Figure 19.** RMSE graph according to N value.

**Table 8.** RMSE of train and test data according to **N** and node value of LSTM algorithm.

| N | RMSE | | Node (N = 5) | RMSE | |
|---|---|---|---|---|---|
| | **Train Data** | **Test Data** | | **Train Data** | **Test Data** |
| 1 | 0.1929 | 0.6444 | 1 | 0.2168 | 0.7171 |
| 2 | 0.1568 | 0.7071 | 2 | 0.1185 | 0.5553 |
| 3 | 0.1431 | 0.6136 | 3 | 0.0916 | 0.5516 |
| 4 | 0.1060 | 0.6012 | 4 | 0.0637 | 0.6410 |
| 5 | 0.0543 | 0.7550 | 5 | 0.0543 | 0.7550 |
| 6 | 0.0485 | 0.8361 | 6 | 0.0394 | 0.6653 |
| 7 | 0.0528 | 0.7110 | 7 | 0.0364 | 0.6743 |
| 8 | 0.0665 | 0.6242 | 8 | 0.0368 | 0.5933 |
| 9 | 0.0520 | 0.7308 | 9 | 0.0258 | 0.6317 |
| 10 | 0.0598 | 0.7560 | 10 | 0.0245 | 0.5752 |
| 11 | 0.0585 | 0.6866 | 11 | 0.0199 | 0.7526 |
| 12 | 0.0597 | 0.7270 | 12 | 0.0158 | 0.6669 |
| 13 | 0.0541 | 0.7097 | 13 | 0.0186 | 0.5825 |
| 14 | 0.0610 | 0.7736 | 14 | 0.0134 | 0.6293 |
| 15 | 0.0578 | 0.6540 | 15 | 0.0110 | 0.6345 |
| 16 | 0.0730 | 0.6968 | 16 | 0.0097 | 0.6246 |
| 17 | 0.0577 | 0.8223 | 17 | 0.0114 | 0.6570 |
| 18 | 0.0625 | 0.8660 | 18 | 0.0075 | 0.6507 |
| 19 | 0.0673 | 0.7243 | 19 | 0.0080 | 0.6499 |

As shown in Figure 19, when N value increases, the error of the learning data decreases. However, N value should be applied properly because N increases the computer's computation. If the value of N is less than 5, the slope of the graph is steep. If it exceeds 5, the slope is gentle. In order to improve training efficiency, we designated N to be 5. We then calculated RMSE.

Figure 20 presents RMSE values of train data and test data according to the number of nodes.

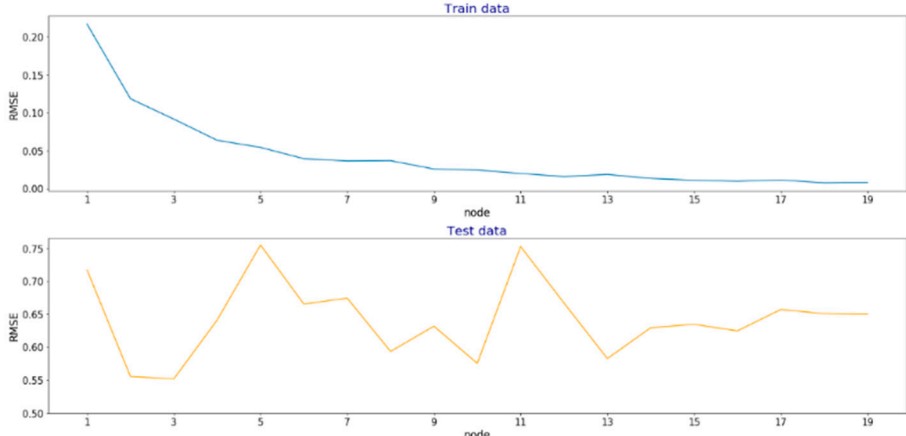

**Figure 20.** RMSE according to node value.

*5.3. CNN*

CNN has the advantage to easily train two-dimensional data. In addition, the number of parameters is small [28]. CNN algorithm can represent weighting value and bias term like Equation (9). The computation processing of MLP is the same as that of CNN. However, CNN has a feature in that its input data are organized by two dimensions.

$$Y_{CNN} = WX + b \tag{9}$$

We created two-dimensional data by designating the column as time flow and the row as variable to be applied to the CNN algorithm in this paper.

Figure 21 shows a data reordering process and a calculation process for the CNN algorithm. The table in Figure 21 shows that when data are merged, the row in table is time flow and the column consists of variable and target value.

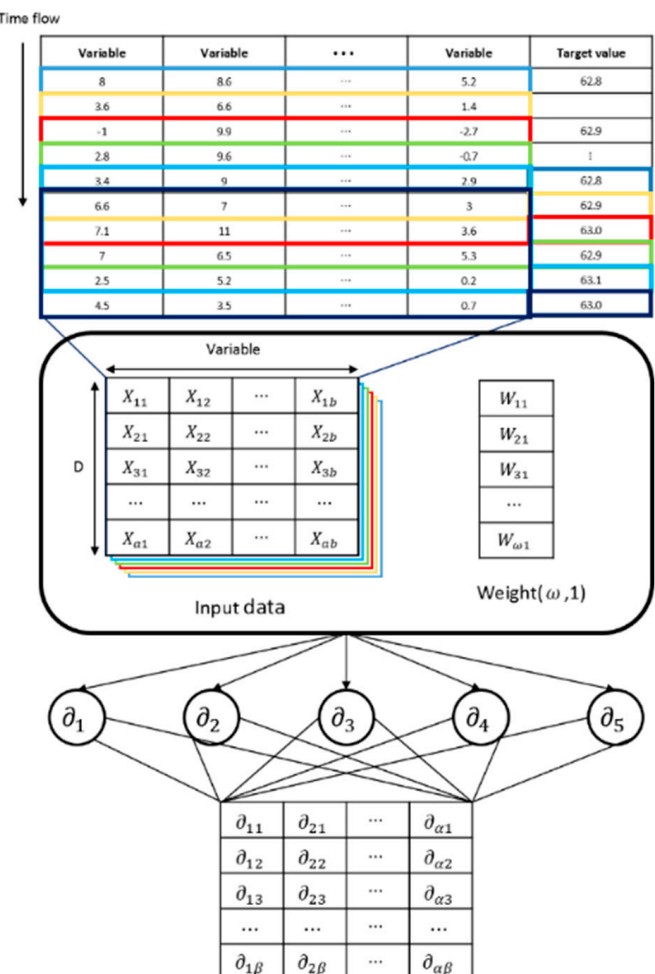

**Figure 21.** Data reordering process and the calculation process for the CNN algorithm.

Input data of CNN had two dimensions. The column was variable and the row was D. Variables were year, month, day, holiday, and *T_temperature*. D was date. Boxes with various colors inside the table were input data. We made two-dimensional data for each one-day stride. Noting that the most empirically optimized value of the N learning course in the LSTM learning course was 5. We designated D as 5, the same value as N.

Target value is the amount of EPT the algorithm that is trying to predict. Since input data used past data, target value data before D-day without past data were not available. Weight stands for *W* for Equation (9), while Weight's shape is designated as ($\omega$, 1). $\partial$ stands for node of convolution layer and $\partial_{\alpha\beta}$ stands for output value of node.

Figure 22 shows a training process for the algorithm. Flatten of Figure 22 changes the output value of the secondary source convolution layer to one dimension. Since the value of target value was one power transaction volume, we designated the number of nodes in the output layer as one.

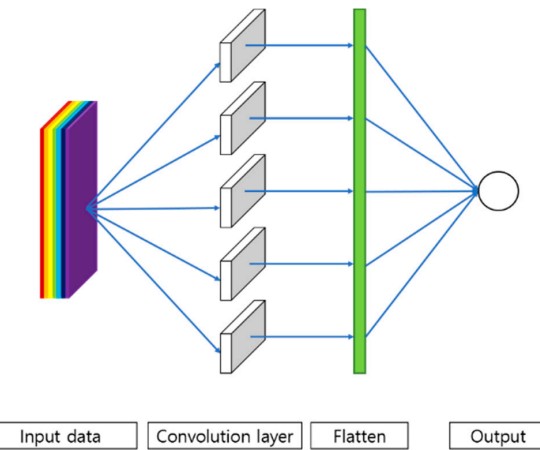

**Figure 22.** Training process of the algorithm.

Table 9 presents RMSE of train data and test data according to the size of D and ω when the number of nodes is five. We could see that the larger the D value, the smaller the error. After this, the error did not change significantly according to D value even when D was greater than 5.

**Table 9.** RMSE according to D and ω.

| Filter (Node = 5) | | RMSE | | Filter (Node = 5) | | RMSE | |
|---|---|---|---|---|---|---|---|
| D | ω | Train Data | Test Data | D | ω | Train Data | Test Data |
| 1 | 1 | 0.7095 | 5.3227 | 6 | 4 | 0.5284 | 3.8690 |
| 2 | 1 | 0.6881 | 5.2076 | 6 | 5 | 0.5273 | 3.8889 |
| 2 | 2 | 0.6787 | 5.1663 | 6 | 6 | 0.5297 | 3.9206 |
| 3 | 1 | 0.6619 | 5.0385 | 7 | 1 | 0.5361 | 3.9313 |
| 3 | 2 | 0.6645 | 5.0485 | 7 | 2 | 0.5433 | 3.9700 |
| 3 | 3 | 0.6668 | 5.0344 | 7 | 3 | 0.5285 | 3.9058 |
| 4 | 1 | 0.6293 | 4.7151 | 7 | 4 | 0.5329 | 3.9407 |
| 4 | 2 | 0.6214 | 4.7711 | 7 | 5 | 0.5246 | 3.8812 |
| 4 | 3 | 0.6216 | 4.6647 | 7 | 6 | 0.5435 | 4.0078 |
| 4 | 4 | 0.6242 | 4.6801 | 7 | 7 | 0.5310 | 3.9299 |
| 5 | 1 | 0.5512 | 4.0790 | 8 | 1 | 0.5323 | 3.9287 |
| 5 | 2 | 0.5500 | 4.0490 | 8 | 2 | 0.5264 | 3.9228 |
| 5 | 3 | 0.5481 | 4.0558 | 8 | 3 | 0.5365 | 3.9635 |
| 5 | 4 | 0.5564 | 4.0907 | 8 | 4 | 0.5208 | 3.8832 |
| 5 | 5 | 0.5474 | 4.0393 | 8 | 5 | 0.5346 | 3.9507 |
| 6 | 1 | 0.5221 | 3.8586 | 8 | 6 | 0.5281 | 3.9112 |
| 6 | 2 | 0.5336 | 3.8973 | 8 | 7 | 0.5305 | 3.9288 |
| 6 | 3 | 0.5204 | 3.8571 | 8 | 8 | 0.5295 | 3.8915 |

Figure 23 shows RMSE according to D and filter size. The higher the RMSE, the darker the red color. The lower the value, the darker the blue color.

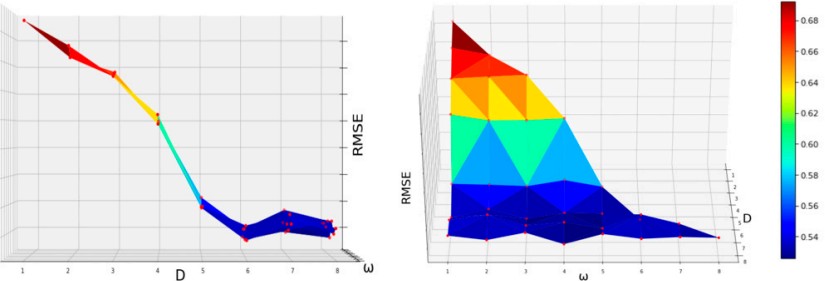

**Figure 23.** RMSE according to D and ω.

*5.4. GRU, SVR, ANFIS*

Chung et al. [29] have proposed GRU using reset gate and update. GSU is a simpler model than LSTM algorithm. When GRU and LSTM are compared, GRU shows better performance than LSTM. In this paper, we trained the LSTM algorithm using input data SVR is an algorithm proposed by Drucker et al. [30]. It is a regression algorithm based on SVM. It can be applied to cases where variables are continuous, unlike SVM.

ANFIS is a combination of neural network and fuzzy theory [31]. This model automatically adjusts the membership function and control rules to fit control object from information of input and output obtained from the control environment using the structure of neural network and train performance.

*5.5. CNN + LSTM*

CNN + LSTM is composed of algorithms of CNN which can easily train 2D data and LSTM which can predict using past information [32]. In this paper, we designated filter size to be 3 and the number of nodes to be 6. We then measured the error using the CNN algorithm. The error did not change significantly according to filter size or the LSTM algorithm as shown in Tables 8 and 9. Figure 24 shows the organization of CNN + LSTM. $\partial$ in Figure 24 represents node of the convolution layer.

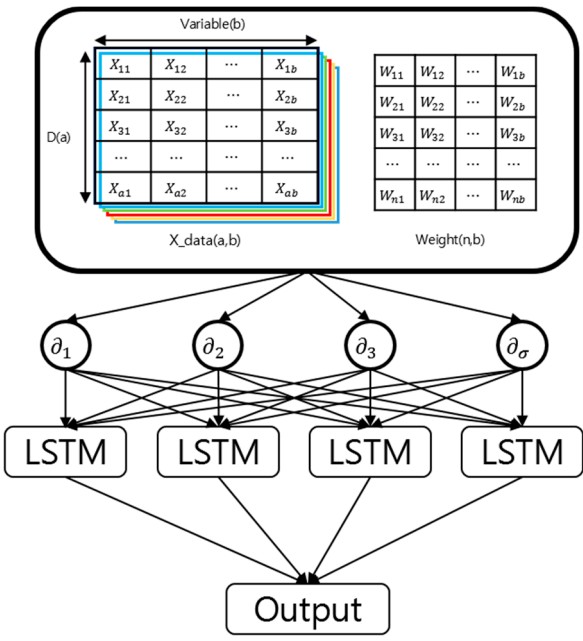

**Figure 24.** Structure of CNN + LSTM.

*5.6. LSTM + CNN*

LSTM is an algorithm that variable enters the hidden layer until before n–day. It ignores the prediction value of the past and outputs prediction value of the current. LSTM + CNN algorithm applies all prediction values of the past and current. It then makes a two-dimensional output value which applies the CNN algorithm.

Like CNN + LSTM in this paper, we designated filter size to be 3 and the number of nodes to be 6. We then measured error using the CNN algorithm in which the error did not change significantly according to filter size or LSTM algorithm as shown in Tables 8 and 9.

Figure 25 presents the LSTM algorithm showing target values of the past and the present. Figure 26 shows the CNN + LSTM structure that connects Figure 25's LSTM algorithm with the CNN algorithm.

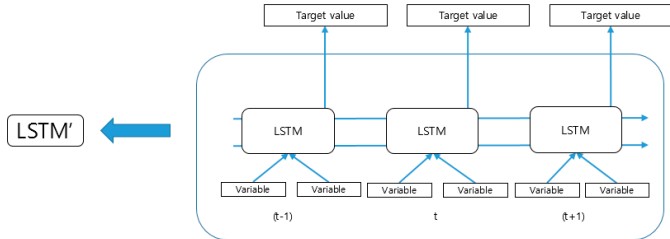

**Figure 25.** LSTM algorithm for selecting all predictions of the past and the present.

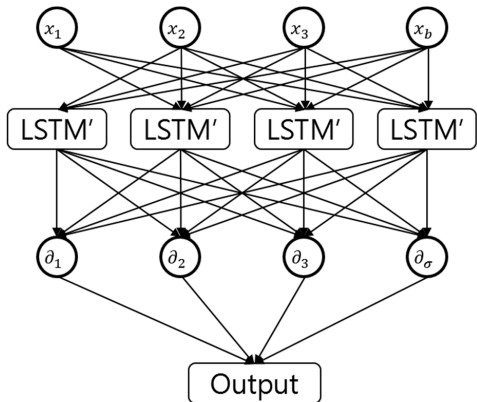

**Figure 26.** Structure of LSTM + CNN.

## 6. Prediction Result

### 6.1. Preprocessing and Non-Processing Comparison

In order to investigate the influence of preprocessing variables, we compared prediction values between non-processing and preprocessing variables. To improve the train performance of the algorithm, we standardized Equation (5). We also measured the precision of performance using data of 2017–2018 rather than data of 2018 as test data.

Figure 27 shows a comparison graph between amounts of EPT and the two predictions for non-preprocess and preprocess. The blue graph and orange graph shows the predicted value and amount of EPT, respectively.

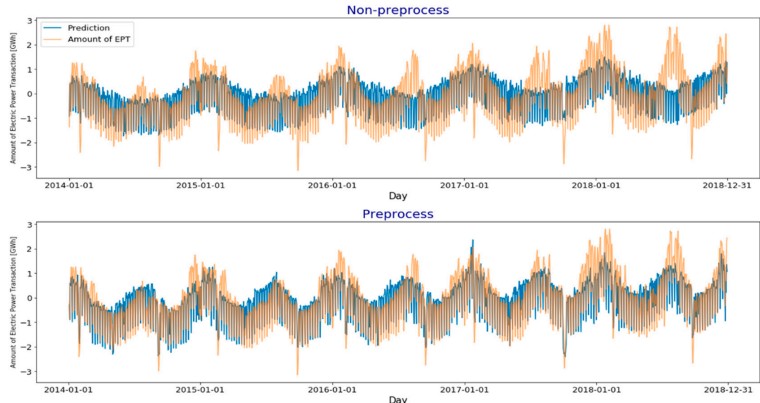

**Figure 27.** Comparison of predictive amount of EPT.

When predicting the amount of EPT using variables without preprocessing, seasonal patterns were not found. The amount of EPT decreased with the same degree on Saturday and Sunday. In addition, the amount of power electric transaction decreased in the same degree on a holiday

and holidays in a row. However, when the amount of EPT was predicted after we added data with preprocessed variables, seasonal patterns, weekend patterns, and special day patterns were observed. These patterns were not completely the same. When measurement errors were compared between preprocessing and not preprocessing, the error was smaller when variables were preprocessed than that without preprocessing. Table 10 shows comparison results of predicted values between preprocessing and non-preprocessing.

**Table 10.** Comparison of prediction results between preprocessing and non-preprocessing of variables.

| Number | Sortation | Pre-Process | | EPT |
| | | Before | After | |
|---|---|---|---|---|
| 1 | Winter → Spring | Decrease | Decrease | Decrease |
| 2 | Spring → Summer | Decrease | Increase | Increase |
| 3 | Summer → Fall | Increase | Decrease | Decrease |
| 4 | Fall → Winter | Increase | Increase | Increase |

### 6.2. Pattern Analysis of Prediction Values

Pattern 1: Seasonal pattern

Figure 28 presents a graph of prediction results for the amount of EPT from January 2017 to December 2017 proposed for sections by applying each algorithm. Blue and gray graph represent predictive amount of EPT of each algorithm and amount of EPT. The prediction performance was determined based on MAPE and RMSE. Results showed that SVR and CNN algorithms had high accuracy. In order to accurately evaluate the performance of each algorithm, we analyzed seasonal, Sunday, and special day patterns for each algorithm from 1 January 2017 to 31 December 2017. Results of pattern analysis revealed that all predicted amount of EPT had seasonal patterns, increasing in summer and winter seasons but decreasing in spring and autumn seasons.

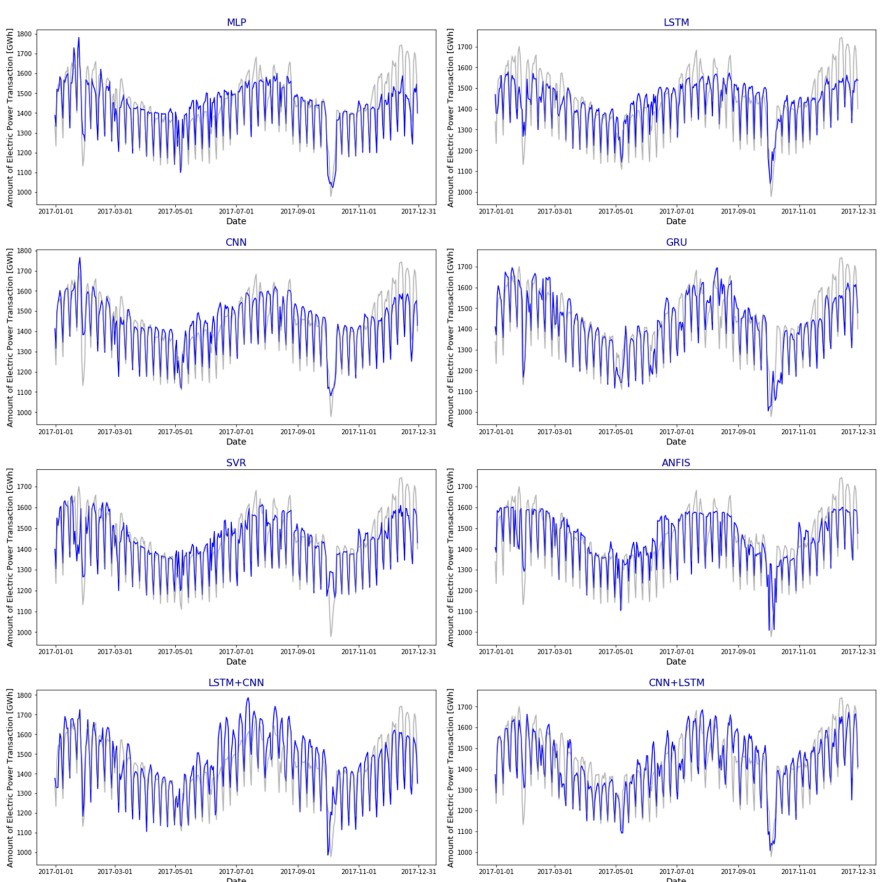

**Figure 28.** Comparison of seasonal patterns with different algorithms.

A large error occurred for the prediction of the amount of EPT in winter season of 2017 using proposed algorithms. However, CNN + LSTM algorithm showed the lowest prediction error among all algorithms.

Pattern 2: Week pattern

Figure 29 shows the amount of EPT from 5 January 2017 to 17 January 2017. Blue and gray graph represent predictive amount of EPT of each algorithm and amount of EPT. The green vertical line and the red vertical line represent Saturday and Sunday, respectively. Week pattern showed that variables of Equation (2) were well applied to all algorithms presented in Section 5.

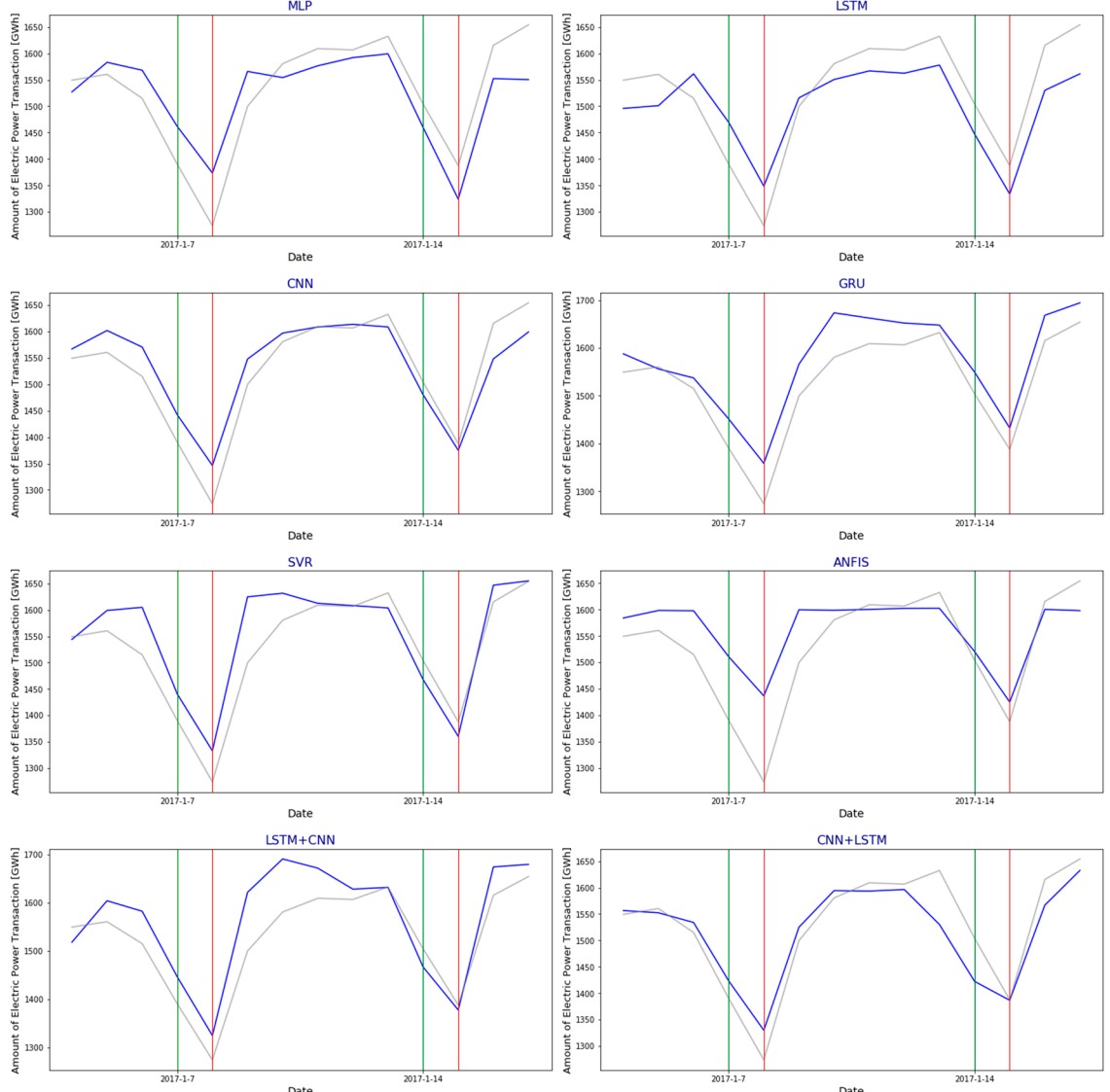

**Figure 29.** Comparison of week patterns with different algorithms.

Pattern 3: Special day pattern

Figure 30 shows the amount of EPT from 3 June 2017 to 9 June 2017. Blue and gray graph represent predictive amount of EPT of each algorithm and amount of EPT. The red vertical line represents a special day of Memorial Day. MLP, CNN, SVR, and ANFIS algorithms predicted that the amount of EPT would decreasing during this period.

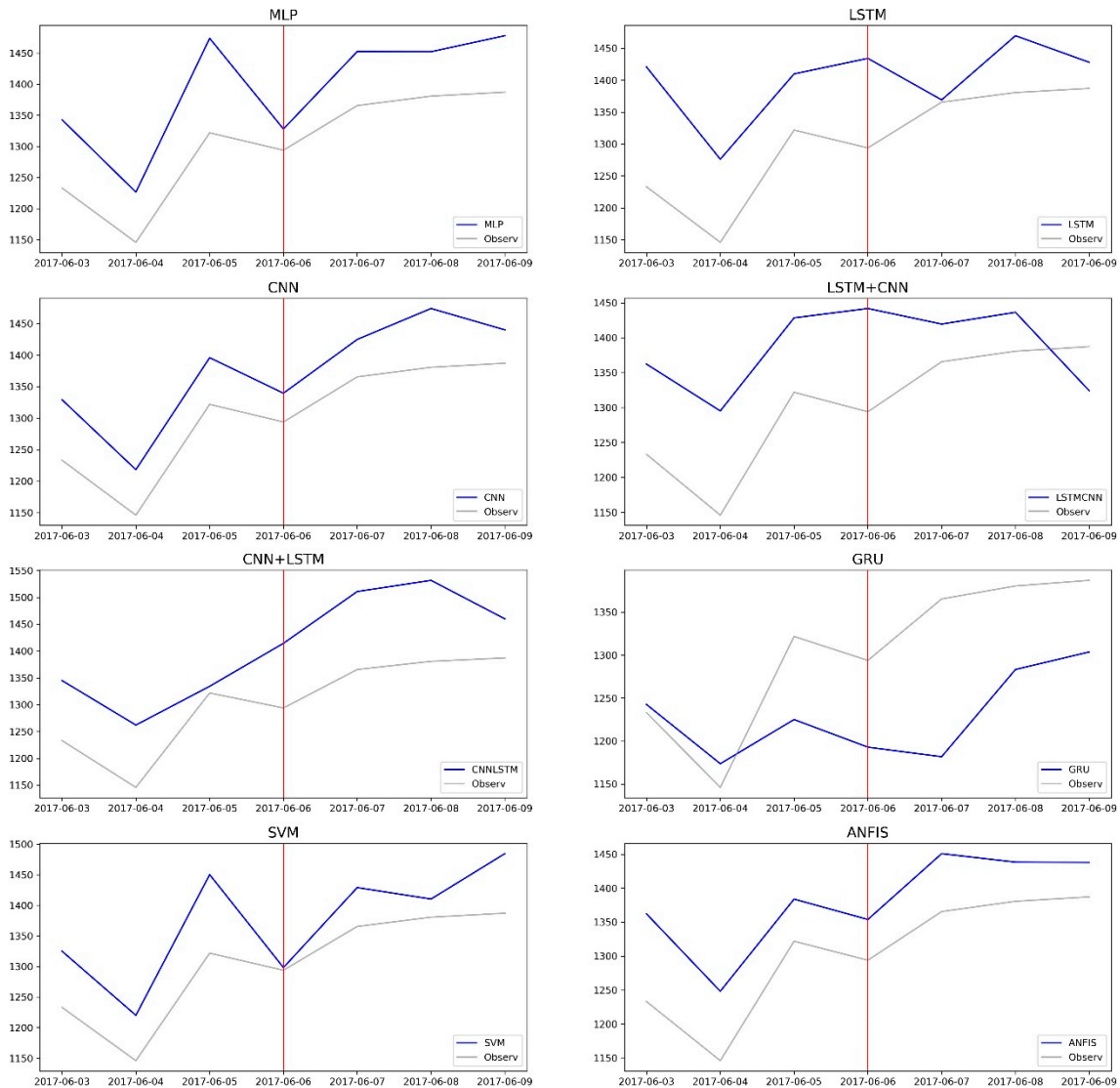

**Figure 30.** Comparison of special day patterns with different algorithms.

Pattern 4: Special day (holiday season) pattern

Figure 31 shows the amount of EPT from 25 September 2017 to 13 October 2017. Blue and gray graph represent predictive amount of EPT of each algorithm and amount of EPT. Red vertical lines represent starting day and ending day of holidays in a row. Holidays in row of October 2017 continued for 10 days, including Armed Forces Day, temporary holidays, opening days, Thanksgiving Day, alternative holidays, and Hangul Day. During the holiday season, four algorithms (MLP, CNN, LSTM, and CNN + LSTM) showed prediction patterns consistent with the holiday pattern displayed in Section 4.3. However, the other four algorithms showed prediction patterns inconsistent with the holiday pattern as the decrease in holidays at the middle point was predicted to be lower than the decrease in the beginning day or the last day.

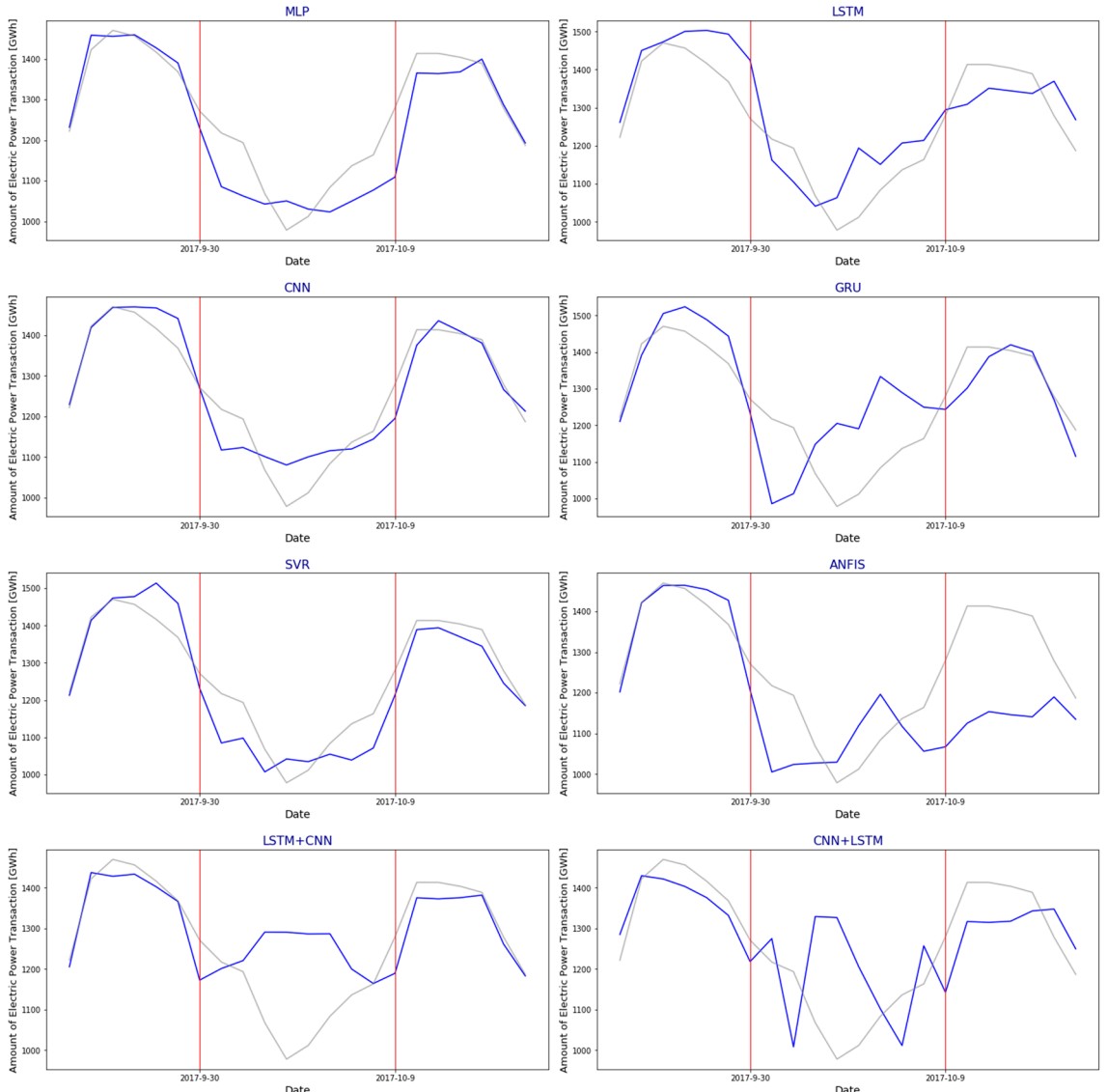

**Figure 31.** Comparison of special day (holiday season) patterns with different algorithms.

Table 11 shows RMSE and MAPE results for the prediction of the amount of EPT from 1 January 2017 to 31 December 2017 using eight algorithms tested in this study, when we apply 60% of train data and 40% of test data. As a result, CNN algorithm showed the best prediction.

**Table 11.** RMSE and MAPE Results of Predicted Values for 1 Year.

|            | RMSE [GWh] | MAPE [%] |
|------------|------------|----------|
| MLP        | 74.720     | 3.970    |
| LSTM       | 82.954     | 4.680    |
| CNN        | 67.350     | 3.513    |
| GRU        | 74.744     | 3.952    |
| SVR        | 71.106     | 3.577    |
| ANFIS      | 75.128     | 4.046    |
| LSTM + CNN | 86.183     | 4.681    |
| CNN + LSTM | 72.840     | 3.975    |

Table 12 also shows RMSE and MAPE results for the prediction of the amount of EPT of 2017 for validation and 2018 for test using eight algorithms tested in this study. As a result, CNN algorithm also showed the best prediction.

**Table 12.** RMSE and MAPE Results of Predicted Values for 2 Year.

| | RMSE [GWh] | | MAPE [%] | |
|---|---|---|---|---|
| | Validation | Test | Validation | Test |
| MLP | 74.720 | 86.424 | 3.970 | 4.220 |
| LSTM | 82.954 | 89.590 | 4.680 | 4.996 |
| CNN | 67.350 | 79.785 | 3.513 | 3.932 |
| GRU | 74.744 | 85.106 | 3.952 | 4.459 |
| SVR | 71.106 | 110.996 | 3.577 | 5.833 |
| ANFIS | 75.128 | 115.571 | 4.046 | 6.215 |
| LSTM + CNN | 86.183 | 81.226 | 4.681 | 10.813 |
| CNN + LSTM | 72.840 | 154.404 | 3.975 | 8.746 |

## 7. Conclusions

In this paper, we investigated seasonal, Sunday, and special day patterns of the amount of EPT in the past to predict the amount of EPT in South Korea. We selected date, holiday, and temperature data. Korean holiday was acquired on the day stipulated in the "Regulations on Public Holidays of Government Offices" by the Presidential Decree of Korea. It was presented with a binary digit.

Since the acquired amount of EPT was the amount of total EPT in South Korea, we designated three large cities in each region and acquired their temperature data. In order to use variables that could be used as references on 31 December 2016, we used temperature data of the prior year.

For example, if we want to predict the amount of EPT in 2017, we used temperature data in 2016. In addition, in order to improve the correlation between variables and amount of EPT, we preprocessed variables. We found differences when we compared predicted values without preprocessing of variables and those with preprocessing of variables.

We tested various algorithms such as MLP, LSTM, CNN, GRU, SVR, ANFIS, CNN + LSTM, and LSTM + CNN to predict the amount of EPT for 2017 of South Korea. When we measured errors for the predicted amount of EPT in 2017 with various proposed algorithms, the CNN algorithm with six nodes and three filter sizes showed the lowest prediction error (67.35 for RMSE and 3.513% for MAPE).

We found that the accuracy of prediction using CNN was the highest among all algorithms for a weekly pattern. However, CNN could not predict the trend of annual amount of EPT in 2017 of South Korea. The prediction error using CNN was large from 1 November 2017.

In the future studies, it is necessary to add variables with characteristics of unforeseen patterns. In this study, we used temperature variable from one year ago. We also need temperature data after one year as variable. To do this, prediction of future temperature, humidity, wind speed and atmosphere pressure are required.

**Author Contributions:** G.B. and Y.B. conceived and designed the conceptualization, G.B. performed the computer simulation and Y.B. analyzed the data and algorithms; G.B. wrote the paper. Both authors have read and agreed to the published version of the manuscript.

**Funding:** This research was supported by Chonnam National University (Smart Plant Reliability Center) grant funded by the Ministry of Education (2020R1A6C101B197).

**Conflicts of Interest:** The authors declare no conflict of interest.

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
