# Peer review of "Predicting the Amount of Electric Power Transaction Using Deep Learning Methods"

_energies, doi:10.3390/en13246649_

Round 1

Reviewer 1 Report

Authors are strongly recommended to get a professional proofread as the manuscript is now very hard to understand. Specifically, abstract is very difficult to follow and the reviewer is afraid that the authors use some words that are not professional but would be avoided in technical papers. For the above reason, the innovation of the paper is not clear. Making sure texts are understandable is indispensable for a good paper.

Author Response

Reviewer1

(Comment) Authors are strongly recommended to get a professional proofread as the manuscript is now very hard to understand. Specifically, abstract is very difficult to follow and the reviewer is afraid that the authors use some words that are not professional but would be avoided in technical papers. For the above reason, the innovation of the paper is not clear. Making sure texts are understandable is indispensable for a good paper.

(Answer) Thank you for your comment.

We trimmed the abstract which is represented as red line.

We also revised some item in the main context.  

Reviewer 2 Report

Strengths

In this contribution the authors use some classical AI algorithms like: MLP, CNN, GRU, SVR etc. They are flexible, robust and used for predictions. Very important in terms of validating the results is the fact that the authors apply the procedure of root-mean-square error (RMSE) in order to measure the differences between predicted and the observed values and of the mean absolute percentage error (MAPE) in order to evaluate how accurate the forecast system operate.

The anticipatory/predictive character of the proposed algorithm is confirmed by the fact that in this paper the predicted start point is placed in the future of used variable field, not like described in other publications, when the predicted start point is older than used variable.

The topic actuality is highlighted by the fact that the publications on the prediction of electric power consumption has been steadily increasing for last years, from 8 in 1999 to 148 in 2018.

Weaknesses

The collected data in 2018 is present both in the test phase (learning/train data sets from 2018), in order to measure the prediction accuracy and in the prediction phase (test), in order to measure the algorithm errors. In this situation, it is recommended to extend the data capture for 2019, to avoid using the same data set in both phases, test and train.

The authors divide the collected data into two categories: learning and test data at a ratio of 70% to 30%.  although a splitting method, 60-70% training, 10% validation and at least20% testing, are usually practiced. In order to increase the traceability of this methodology, please explain better how you distributed the collected data.

The authors used eight variables including prior work. The use of "Prior work (PW)" variables can introduce a "bias" in the new prediction model. As a result, a small error introduced in the PW, can propagate in the model. Did you plan to eliminate this possible error?

Recommendations before editing

Regarding tab.1, please differentiate between the input variables for LTLF and MTLF (range of weeks or even months).

Regarding table 2, we recommend inserting some numerical data (reference values) in the last “description” column, in order to make possible a performances comparison of the previous methods.

MAPE about 3% is a relevant and accurate indicator (Tab. 11). In this context, clarify please the used MU (measurement unit) assigned by RMSE (67.350 is it the square difference between successive predictions / is this value normalized?)

In fig. 3, 4, and 6 please specify what time interval was used for amount of EPT, on the vertical axis (GWh in an hour / day / week)?

Figure 7 shows temperatures from January 01, 2013 to December 31, 2017 but in diagram is extended until 2018. Please adapt it.

Future research

In order to increase the performance of the proposed AEPT algorithm, in the next research it is to be considered other meteorological data: humidity, wind speed and atm. pressure.

The last suggestion for an even more accurate correlation of the results, the input variables in the model should be put in correspondence with the local energy mixture (the Korean energy mix).

Author Response

Reviewer 2

Weaknesses

  1. (Comment) The collected data in 2018 is present both in the test phase (learning/train data sets from 2018), in order to measure the prediction accuracy and in the prediction phase (test), in order to measure the algorithm errors. In this situation, it is recommended to extend the data capture for 2019, to avoid using the same data set in both phases, test and train.

(Answer) Thank you for your comment. We wrote it in line 491 to 494 following as:

“To measure the performance of each algorithm, we divided data into learning (or training) data and test data. Learning data accounted for 60% of data from 2014 to 2016 and test data accounted for 40% of data from 2017 to 2018. We trained the algorithm with learning data and measured the performance of algorithm with test data.”

First we never used learning/test data of data of 2018. We used learning data that is used for 60% of data from 2014 to 2016, and test data accounted for 40% of data from 2017 to 2018. Thus data of 2018 that we used is only test data.

Second, reviewer recommended data capture until 2019. We completely agree with reviewer’s opinion. However, Korea Power Exchange (KPX) and Korea Electric Power Company (KEPCO) never release the power transaction data of 2019. Thus we cannot extend the data until 2019.

Third, due to limitation of getting the data for amount of electrical power transaction, we cannot extend any more data.  

  1. (Comment) The authors divide the collected data into two categories: learning and test data at a ratio of 70% to 30%. Although a splitting method, 60-70% training, 10% validation and at least 20% testing, are usually practiced. In order to increase the traceability of this methodology, please explain better how you distributed the collected data.

(Answer) Thank you for your comment.

Generally, we can use learning data, validation data, and test data to prove performance of the model. However, this paper is not validation data because the use of this data is nearly the same with the test data.

The learning data is used to learn the learning model. However, the validation and test data is used to evaluate the model, not to learn the model.

If we designate the validation, the ratio of learning data and test data will reduce. If the number of learning data is reduced, the model cannot learn to that extent. If the number of test data is reduced, result of prediction performance of model will get worse.

Even though, the result will get worse, we added RMSE and MAPE results for the prediction of the amount of electric power transaction 2017 for validation and 2018 for test in table 12, according to reviewer’s recommendation.

  1. (Comment) The authors used eight variables including prior work. The use of "Prior work (PW)" variables can introduce a "bias" in the new prediction model. As a result, a small error introduced in the PW, can propagate in the model. Did you plan to eliminate this possible error?

(Answer) I think the comment of reviewer is right. The reason of "Prior work (PW)", or pre-processing of data is to reduce the error. Typically, error size of before pre-processing is greater than the error size of after pre-processing, we think “bias” is already reflected when we pre-processing data for temperature and holiday.

Recommendations before editing

  1. (Comment) Regarding tab.1, please differentiate between the input variables for LTLF and MTLF (range of weeks or even months).

(Answer) We changed which is represented as red line.

  1. (Comment) MAPE about 3% is a relevant and accurate indicator (Tab. 11). In this context, clarify please the used MU (measurement unit) assigned by RMSE (67.350 is it the square difference between successive predictions / is this value normalized?).

(Answer) We insert the measurement unit which is represented as red line.

We standardize data which is used in learning and then we learned the predicted amount of electric power transaction.

We do inverse standardization to measure real error. From Fig. 28 to Fig. 31 show the predicted value through inverse standardization. Table 11 and Table 12 represents the error between inverse standardization value and real amount of electric power transaction.

  1. (Comment) In fig. 3, 4, and 6 please specify what time interval was used for amount of EPT, on the vertical axis (GWh in an hour / day / week)?

(Answer) We changed “day” instead of “date”.

  1. (Comment) Figure 7 shows temperatures from January 01, 2013 to December 31, 2017 but in diagram is extended until 2018. Please adapt it.

(Answer) We adapted it.

Future research

(Comment) In order to increase the performance of the proposed AEPT algorithm, in the next research it is to be considered other meteorological data: humidity, wind speed and atm. pressure.

The last suggestion for an even more accurate correlation of the results, the input variables in the model should be put in correspondence with the local energy mixture (the Korean energy mix).

(Answer) Thank you for your comment. We will apply your suggestion for next research.

Reviewer 3 Report

Fırst of all, thank you to let me read your research. At first glance, I can say that you have a quite long abstract, I would recommend you to keep it a bit short. The research is about the prediction of electric power transaction based on seasons, holidays, and special days using deep learning algorithms.  There is an extensive study with full of knowledge. The study is well presented. There is only a bit of a problem with the grammar and positioning of the shapes. Using MLP, LSTM, CNN, GRU, SVR, ANFIS, CNN+LSTM, and LSTM+CNN and having only CNN with the lowest error results(67.35 for RMSE and 3.513% for MAPE) is quite interesting. I was expecting LSTM or LSTM+CNN to be the winner. Finally, please go ahead and fix the following issues.

  • Table 1 should be on one page not seperated into two pages.
  • Figure 14 is not readable, please revise.
  • line 646  and 652 font size is not the same as the rest of the article, please correct.

Author Response

Reviewer3.

  1. (Comment) Fırst of all, thank you to let me read your research. At first glance, I can say that you have a quite long abstract, I would recommend you to keep it a bit short.

(Answer) Thank you for your comment. We trimmed the abstract which represented as red line.

  1. (Comment) The research is about the prediction of electric power transaction based on seasons, holidays, and special days using deep learning algorithms.  There is an extensive study with full of knowledge. The study is well presented. There is only a bit of a problem with the grammar and positioning of the shapes. Using MLP, LSTM, CNN, GRU, SVR, ANFIS, CNN+LSTM, and LSTM+CNN and having only CNN with the lowest error results (67.35 for RMSE and 3.513% for MAPE) is quite interesting. I was expecting LSTM or LSTM+CNN to be the winner. Finally, please go ahead and fix the following issues.
  2. (Comment) Table 1 should be on one page not separated into two pages.

(Answer) We revised as one page.

  1. (Comment) Figure 14 is not readable, please revise.

(Answer) We changed.

  1. (Comment) line 646 and 652 font size is not the same as the rest of the article, please correct.

(Answer) we changed.

Round 2

Reviewer 1 Report

The authors have addressed all the Reviewer commends. Please go through a final proofread prior to publication.